# Food amyloid fibrils are safe nutrition ingredients based on in-vitro and in-vivo assessment

Dan Xu[1,2,8], Jiangtao Zhou ®[2,8] ✉, Wei Long Soon[2,3], Ines Kutzli[2], Adrian Molière[4], Sabine Diedrich[2], Milad Radiom ®[2,5], Stephan Handschin[6], Bing Li[1], Lin Li[1], Shana J. Sturla[2], Collin Y. Ewald[4] & Raffaele Mezzenga ®[2,7] ✉

Food protein amyloid fibrils have superior technological, nutritional, sensorial, and physical properties compared to native monomers, but there is as yet insufficient understanding of their digestive fate and safety for wide consumption. By combining SDS-PAGE, ELISA, fluorescence, AFM, MALDI-MS, CD, microfluidics, and SAXS techniques for the characterization of β-lactoglobulin and lysozyme amyloid fibrils subjected to in-vitro gastrointestinal digestion, here we show that either no noticeable conformational differences exist between amyloid aggregates and their monomer counterparts after the gastrointestinal digestion process (as in β-lactoglobulin), or that amyloid fibrils are digested significantly better than monomers (as in lysozyme). Moreover, in-vitro exposure of human cell lines and in-vivo studies with *C. elegans* and mouse models, indicate that the digested fibrils present no observable cytotoxicity, physiological abnormalities in health-span, nor accumulation of fibril-induced plaques in brain nor other organs. These extensive in-vitro and in-vivo studies together suggest that the digested food amyloids are at least equally as safe as those obtained from the digestion of corresponding native monomers, pointing to food amyloid fibrils as potential ingredients for human nutrition.

*Homo sapiens* has been consuming dairy and poultry proteins since at least the Neolithic age, and no concerns exist about possible toxicity of these proteins in their native form, besides occasional allergenicity cases in a very restricted part of the human population. Yet, due to safety concerns regarding their consumption in a denatured form, specifically aggregated into amyloid fibrils, amyloid dairy and poultry proteins are not widely used. In fact, safety concerns associated with the amyloid status of aggregated proteins in general prevent use of

food amyloid fibrils in foods, independent of the original food source from which the native protein precursor is extracted. From a structural perspective, amyloid fibrils are fibrillar aggregates of proteins characterized by a robust amyloid core of a highly-ordered cross-β architecture[1]. Protein amyloid fibrils were discovered first as pathological entities, associated with a variety of human disorders, including certain devastating neurodegenerative diseases, such as Alzheimer's, Parkinson's and Huntington's diseases, as well as many other systemic

[1]School of Food Science and Engineering, Guangdong Province Key Laboratory for Green Processing of Natural Products and Product Safety, Engineering Research Center of Starch and Plant Protein Deep Processing, Ministry of Education, South China University of Technology, 381 Wushan RoadTianhe District Guangzhou 510640, China. [2]Institute of Food, Nutrition and Health (IFNH), Department of Health Sciences and Technology (HEST), ETH Zurich, Zürich 8092, Switzerland. [3]Center for Sustainable Materials (SusMat), School of Materials Science and Engineering, Nanyang Technological University, Singapore 639798, Singapore. [4]Institute of Translational Medicine, Department of Health Sciences and Technology (HEST), ETH Zurich, Schwerzenbach, Switzerland. [5]Laboratory of Food Immunology, Institute of Food, Nutrition and Health, ETH Zürich, Zürich, Switzerland. [6]Scientific Center for Optical and Electron Microscopy (ScopeM), ETH Zurich, Otto-Stern-Weg 3, 8093 Zurich, Switzerland. [7]Department of Materials, ETH Zurich, Zürich 8092, Switzerland. [8]These authors contributed equally: Dan Xu, Jiangtao Zhou. ✉e-mail: jiangtao.zhou@hest.ethz.ch; raffaele.mezzenga@hest.ethz.ch

amyloidoses[2,3]. However, an increasingly thorough understanding of amyloid aggregation suggests that these proteinaceous nanofibrils are not limited to pathologies, and the fibrillization propensity could constitute a generic property of many, if not all, existing proteins[4].

In recent years, numerous examples of non-pathological amyloids have been found in various organisms, including humans[2,5]. The non-disease-related functional amyloids arise from many different proteins, including spider silk[6], *E. coli* Curli fibrils[7] and human receptor-interacting protein kinase[8]. These amyloids play important physiological roles in living organisms, and in some cases even in humans, such as endocrine hormone peptides, which are stored in secretory granules in the form of amyloids[9]. More recently, artificial amyloids from natural proteins or synthetic polypeptides were intensively studied, and employed in diverse advanced nanotechnologies[10–12]. Among them, food proteins are an ideal source for manufacturing artificial amyloid materials in-vitro because of the non-toxic nature of the protein precursor, their natural abundance, and low cost[4,13].

Due to their remarkable physiochemical and nanomechanical properties, food protein amyloids have been used as biomaterials with a wide variety of superior functionalities, ranging from biomedicine and environmental science to material science and nanotechnology[12,14]. These food protein amyloids, which include milk β-lactoglobulin[15], egg-white lysozyme[16], ovalbumin[17], oat[18] and other proteins[19–21] could offer substantial potential for food and pharmaceutical industries[4,22]. Indeed, due to their rich functional groups and high surface-to-volume ratio, amyloid fibrils are capable of carrying nutrients for delivery[23], and are utilized for encapsulation of drug-like molecules in microgels or gel shells[24]. Additionally, biocompatible hybrid biomaterials based on amyloid fibrils could be promising in antitumor therapy[25], anti-obesity treatments[26], and other biomedical applications[27]. However, in stark contrast to their remarkable success in many technologies and applications, the implementation of food protein amyloids into the human diet has long been hampered, mostly due to the extrapolation to the pathological amyloid class[4]. Since no systemic or pathological amyloidosis associated to food protein amyloids is reported to date, current concerns about their use in human nutrition remain associated with either the general nanotoxicity effects, or more specifically, the possibility that associated aggregates reach the bloodstream and nucleate the formation of pathological amyloids by cross-seeding[4,13]. Therefore, the most critical current question concerns whether amyloid nuclei resist gastrointestinal digestion, whether the digested amyloids are cytotoxic, and whether the digested remaining counterparts cross epithelium into the bloodstream and induce cross-seeded pathological aggregates.

Here, we report a comprehensive investigation of the biological fate of amyloid fibrils from β-lactoglobulin (β-lg) through a simulated in-vitro gastrointestinal digestion, in comparison with the native β-lg protein (Fig. 1a). This milk-derived β-lg is one of the most intensively studied food proteins for amyloid formation, and it was therefore chosen as our main focus in this digestion study; we also expand these key results on in-vitro gastrointestinal digestion to hen egg white lysozyme (Lys) amyloids to evaluate the generality of our findings. Our results show that β-lg amyloids significantly degrade after gastric digestion and can be broken down in the following intestinal digestion, resulting in unfolded oligopeptides, indistinguishable from the products of in-vitro digested β-lg monomer. Generalization to lysozyme amyloids shows not only that the residual digested peptides are unfolded oligopeptides but even that the amyloid digestion is more extensive than the corresponding monomers. These results strongly support that physiologically, β-lg and lysozyme amyloid fibrils would degrade into unfolded amino acids polypeptides, which are, therefore, at least equally safe (β-lg) if not safer (lysozyme), than those obtained from digested native monomers. We further consolidate these results by evaluating the impact of the digestates on the viability of human intestinal epithelium cells and on in-vivo studies carried out on

*C. elegans* and mice. The results further support that the digested amyloids present no detectable cytotoxicity in human cells, nor adverse effects on movements during aging on *C. elegans*, nor any sign of cross-seeding to pathological amyloids in the investigated animal models upon long-term administration. In brief, these in-vitro and in-vivo results together suggest food amyloid fibrils as a possible safe nutritional ingredient in human health applications.

## Results

### SDS-PAGE and ELISA assay

We first followed the dynamic proteolysis digestions of β-lg monomers and amyloid fibrils with SDS-PAGE. The monomeric β-lg mainly remained intact with a molecular weight of 18 kDa during gastric digestion (Fig. 1b) with only a small proportion of digested β-lg fragmented into two segments. This indicates the protease resistance of β-lg monomer while exposed to pepsin at the acidic condition, and this observed resistance may be due to the hydrophobic globular structure of β-lg monomer at low pH[28,29]. The indigestibility of β-lg monomers was also reflected in its high allergenicity, as indicated by ELISA assay (Fig. 1d). This is a technique for detecting and quantifying target macromolecules such as peptides, proteins or antibodies, as antigens via specific immobilization with an antibody linked to a reporter enzyme. Here, we applied ELISA to detect the several motifs identified to be IgE-binding epitopes associated more frequently with milk allergy[30,31] and have been suggested to contain pepsin cleavage sites. These motifs, which are reported to be part of the hydrophobic cluster in the core of β-lg under acidic pH[32,33], render themselves inaccessible to pepsin hydrolysis and could potentially be the trigger of allergenic responses upon entering the intestinal tract. This resistance to digestion was also observed in the appearance of only a small proportion of short polypeptides after gastric digestion, in the case of β-lg monomers. Interestingly, we found a different digestion behavior on β-lg amyloid fibrils (Fig. 1c), observed to be digested into polypeptides of less than 10 kDa within the first few minutes after pepsin introduction, during which their allergenicity dropped significantly (Fig. 1e), hinting that the allergenic epitopes could have been more easily accessible to pepsin. Similarities of the peptide sequences from β-lg amyloid fibrils after fibrillization[34] with some of the IgE-binding epitopes from earlier studies highly support that these allergenic motifs are more susceptible to pepsin hydrolysis in the amyloid form due to surface presentation[30,31,33]. With the introduction of pancreatin in the intestinal phase, both the remaining undigested β-lg monomer and polypeptides from amyloid fibrils from the gastric phase began to hydrolyze into shorter polypeptides of less than 5 kDa (Fig. 1c). An SDS-PAGE gel with a higher resolution at lower molecular weights is shown in Supplementary Fig. 1. Their allergenicity was greatly reduced, as cleavage of allergenic motifs resulted from the opening of the compact structure of β-lg monomer due to the increase in pH[35], while remaining motifs left on amyloid fibrils uncleaved by pepsin were further degraded by pancreatin. In essence, this shows that the previously undigestible β-lg monomer became much more digestible after fibrilization starting from the gastric tract. This observation was further supported and strengthened by the study of the other system considered, lysozyme, where lysozyme amyloid fibrils were also observed to have improved digestibility and low allergenicity compared to its monomeric form, not only after the gastric tract, but even after the complete gastrointestinal digestion (Supplementary Figs. 2 and 3).

### Florescence assay and CD spectroscopy

To investigate the possibility that amyloid core fragments withstand gastrointestinal digestion, thiazole orange (TO) fluorescence assay was applied to track the enzymatic degradation kinetics of β-sheet content in the monomeric and amyloid solution during digestion (Fig. 1f–g). As seen in Fig. 1f, TO intensity of β-lg monomer exhibited a relatively slight decrease of 20% in the gastric phase as expected from their

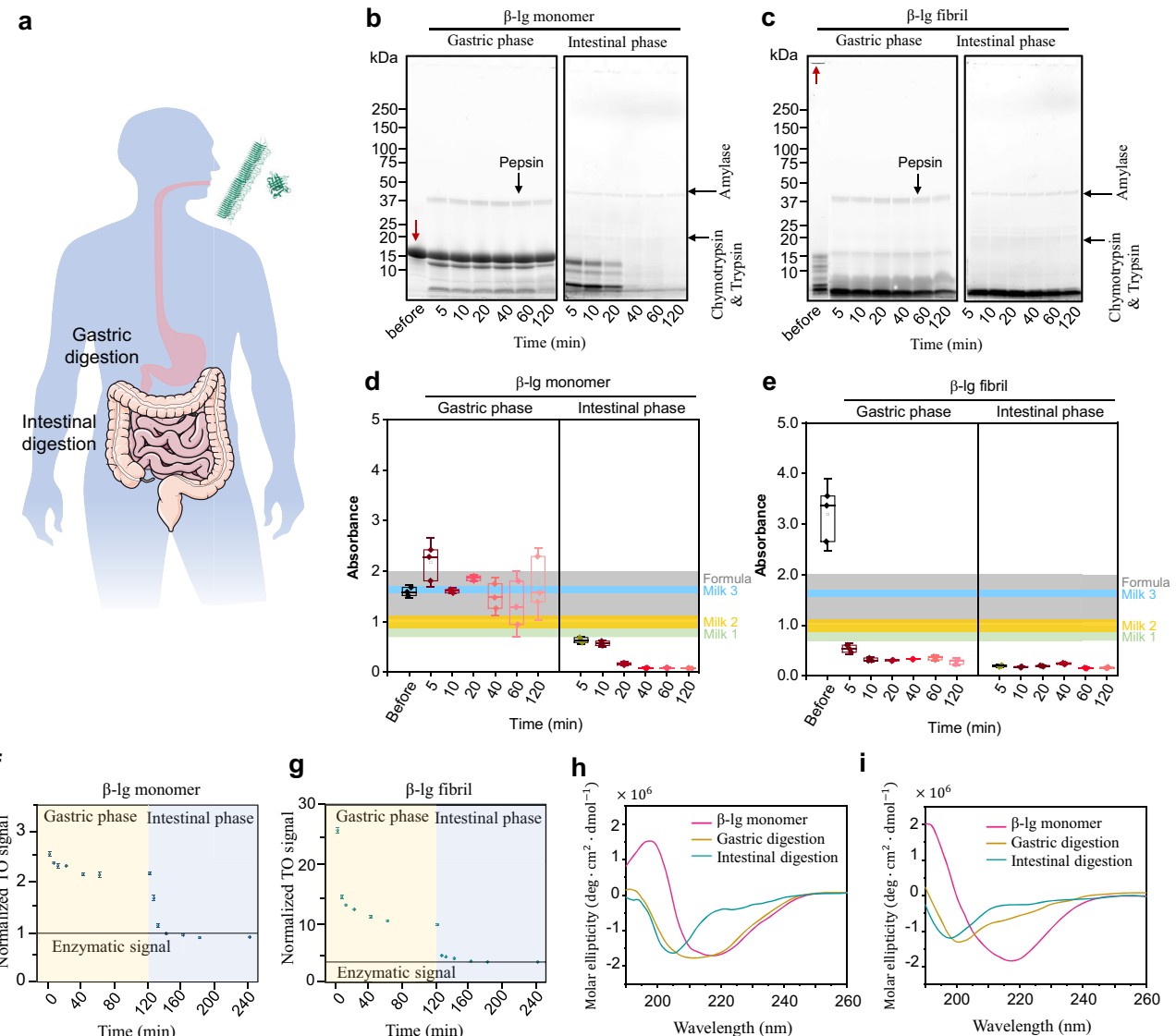

**Fig. 1 | Molecular and structural investigation of β-lg monomers and amyloid fibrils during in-vitro gastrointestinal digestion. a** Simulated gastrointestinal digestion phases of interest. This figure was partly generated using Servier Medical Art provided by Servier licensed under a Creative Commons Attribution 3.0 unported license [https://smart.servier.com/how-to-cite-servier-medical-art/]. **b, c** SDS-PAGE of β-lg monomer (**b**) and amyloid fibrils (**c**). The red arrows indicate the β-lg monomer (**b**) and amyloid fibrils (**c**). **d, e** ELISA antigenicity assay of β-lg monomer (**d**) and amyloid fibrils (**e**). The boxes cover the range of 25–75 percentage, the whisker lines and center lines refer to SD and median value respectively and square symbols indicate the mean value. $n = 3$ independent experiments. **f, g** TO fluorescence assay of β-lg monomer (**f**) and amyloid fibrils (**g**). The enzymatic signal refers to background contribution from digestive enzymes. $n = 3$ independent experiments. **h, i** CD spectra of β-lg monomer (**h**) and amyloid fibrils (**i**) before and after digestion.

resistance to pepsin hydrolysis, followed by a steep decay in the first 20 min of intestinal digestion to the background level of enzymatic solution. In contrast, the fluorescence intensity of β-lg amyloid fibril decreased exponentially by 70% in the gastric phase, reaching as low as the background enzymatic signal in the intestinal phase (Fig. 1g). These results were further confirmed by Thioflavin T (ThT) fluorescence assay in Supplementary Fig. 4. A similar trend was also observed for the digestion of lysozyme amyloid fibrils, during which the addition of pepsin in the gastric phase resulted in a nearly 4-fold reduction in signal, pointing at the facile digestion of lysozyme amyloid fibrils (Supplementary Fig. 5). A slower digestion in the intestinal phase might be due to the change of the degraded polypeptide charge states upon the gastric to intestinal pH transition with possible aggregation of these polypeptides.

Bulk technique CD spectroscopy was performed to detect the presence of β-sheet content before and after gastrointestinal

digestion. β-lg monomer consisted of a mixture of β-sheet and α-helix structures as shown by a broad peak ranging from 210 to 220 nm[36] (Fig. 1h), which shifted slightly towards a lower wavelength in the gastric phase due to partial degradation by pepsin and further shifting towards the typical random coil structure in the intestinal phase. β-lg amyloid fibrils showed a spectrum with a large single minimum at 218 nm (Fig. 1i), indicating the predominant β-sheet structure. This absorption dip shifted significantly to 200 nm after gastric digestion and further shifted to 198 nm after intestinal digestion, exhibiting a dominating random coil structures[37]. These findings provide evidence that both β-lg amyloid fibril and β-lg monomer can be decomposed into random coil polypeptides after gastrointestinal enzymatic reaction, with no detectable β-sheet residues in the protein solution, supporting the observations in Fig. 1f, g and suggesting the absence of β-sheet amyloid cores after digestion. The gastrointestinal digestion of lysozyme amyloid fibrils was also shown to be much more efficient

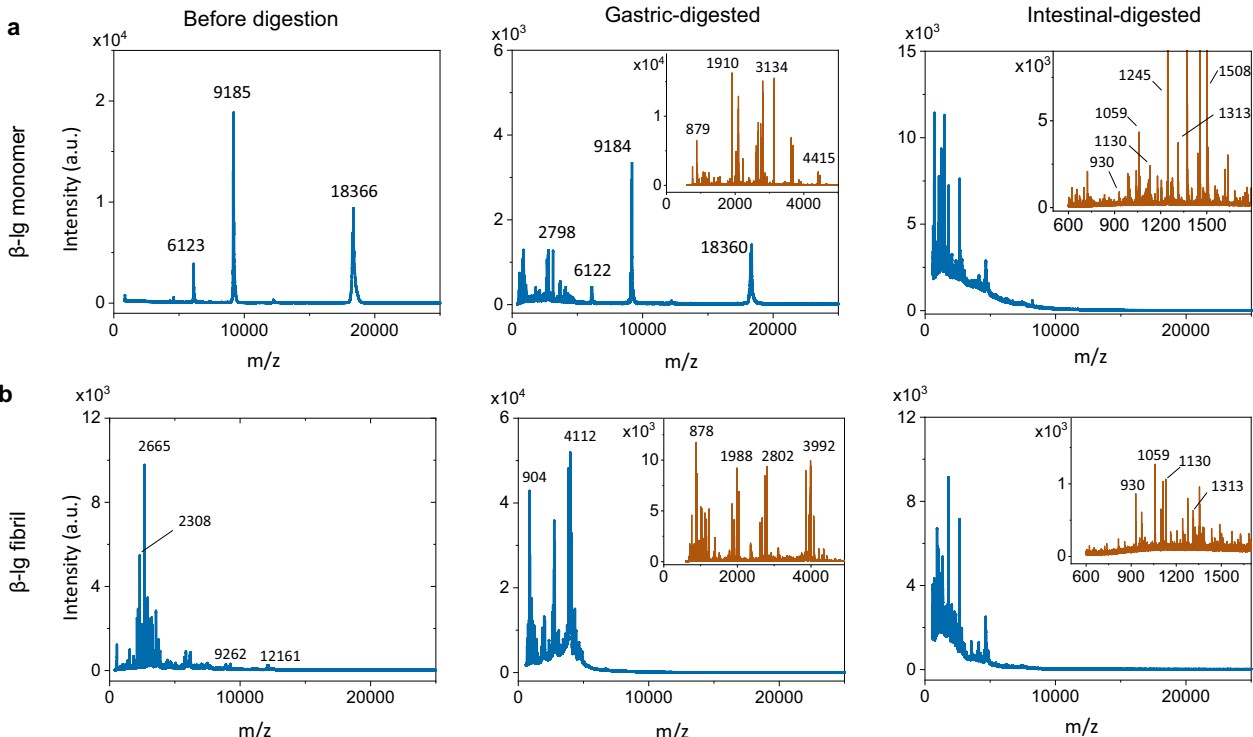

**Fig. 2 | MALDI-MS spectra of β-lg monomer and amyloid fibrils digested products. a, b** β-lg monomer (**a**) and amyloid fibril (**b**) before digestion, after gastric, and intestinal digestion. The a.u. refers to arbitrary units. Full range m/z was acquired in linear mode and insets are in reflector mode.

than that of its monomeric form, exhibiting an obvious transition from the β-sheet peak towards random coil structures while the monomer was observed not to have been as well-digested (Supplementary Fig. 6).

**MALDI-mass spectrometry analysis**

To further determine the composition of digested β-lg monomers and amyloids, matrix-assisted laser desorption/ionization mass spectrometry (MALDI-MS) first was carried out under linear mode detection (Fig. 2). The spectrum of native β-lg monomer showed three main dominant peaks, corresponding to singly ($M_{avg} ≈ 18366$ Da) $[M + H]^+ = (18366 + 1)/1 = 18367$, doubly $[M + 2H]^{2+} = (18366 + 2)/2 = 9184$, and triply charged $[M + 3H]^{3+} = (18366 + 3)/3 = 6123$ full-length β-lg (Fig. 2a). After gastric digestion, we again observed these three peaks of β-lg monomer remaining in addition to partially digested polypeptides below 5 kDa. The inserted plot obtained in reflector mode provides additional details about the spectrum. After intestinal digestion, along with the disappearance of the main β-lg monomer peaks, the distribution of molecular weight shifted towards species of less than 5 kDa, leaving mostly digested oligopeptides of 1–2 kDa corresponding to ~10–20 amino acids. Amyloids after gastric digestion exhibited four main peaks of less than 4 kDa. Specifically, some oligopeptides were digested below 1 kDa, other larger oligopeptides down to 3 kDa or more. Subsequent intestinal digestion of amyloid produced a large number of polypeptides with a molecular weight lower than 5 kDa centered around 1–2 kDa similar to those of β-lg monomer. Similar spectra were obtained after filtering the digested amyloid solution with 10 kDa filters (Supplementary Fig. 7). Interestingly, in the reflector mode spectrum after intestinal digestion we found a total of 100 peaks in the β-lg monomers and only 55 peaks in the β-lg amyloids (see Supplementary Table 1); out of the 55 detected peaks found in digested β-lg amyloid fibrils, 38 were also found on digested β-lg monomers, 15, did not belong to the β-lg sequence (attributed to enzyme autolysis) and only two peaks (at 815.45 Da and

1354.71 Da) were found in the digested fibrils but not the digested monomers. These two peaks can be sequenced into either VRTPEVD, or NENKVLV or VEELKPT for the 815.45 Da peak and YSLAMAASDISLL or QKVAGTWYSLAM for the 1354.71 Da peak. In contrast, in the case of the intestinally digested β-lg monomer we found as many as 22 peaks present only in the monomer sequence (but not in the amyloid sequence) with 40 peaks not belonging to the β-lg sequence (attributed again to enzyme autolysis). This implies that only the two peaks at 815.45 Da and 1354.71 Da are unique to the digested β-lg fibrils, while as many as 37% (22:60) of the fragments are unique to the digested β-lg monomers. However and most importantly, we do find all the possible sequence candidates for these two peaks at 815.45 Da and 1354.71 Da also occurring in larger fragments within the digested β-lg monomers, simply indicating that the digestion of the β-lg amyloids has proceeded to a significantly larger extent than in the case of the β-lg monomers (see Supplementary Table 1), and thus, that residual oligopeptides from digested β-lg fibrils cannot be more toxic than those from the digested homolog β-lg monomers. This conclusion is also reinforced by the digestion behavior of lysozyme−lysozyme monomers were barely digested and remained even after intestinal digestion, whereas lysozyme amyloid fibrils were digested mostly into polypeptides of less than 1 kDa (Supplementary Fig. 8). We should note here that the MALDI-MS spectrometry is carried out at low enzyme content to minimize autolysis of the enzyme and that a more detailed sequencing of the digested peptides is carried out in the in-vivo part where the enzyme is present at physiological conditions (see below for details).

**In-situ microfluidics screening system under polarized light microscopy**

To further understand the kinetics of amyloid fibril digestion, a circulatory microfluidic system was set up for monitoring the dynamic in-situ digestion process of amyloid fibrils under cross-polarized light microscope (Fig. 3a). Amyloid fibril solution was incubated in a water bath at 37 °C with constant shaking. Amyloid fibrils in the solution were

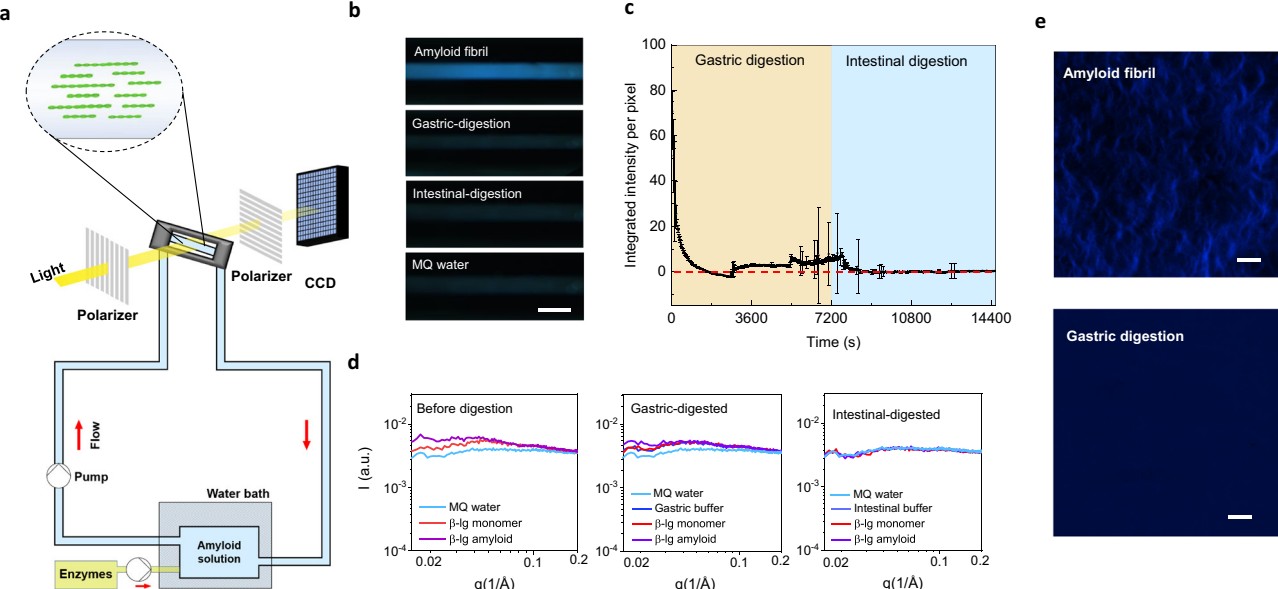

**Fig. 3 | In-situ screening of the β-lg amyloid fibril degradation behavior during digestion. a** Schematic of microfluidic polarized light microscopy setup. **b** Polarized light microscopy images of amyloid fibrils during digestion. The result is verified by two independent tests. Scale bar is 1 mm. **c** Intensity per pixel over time. Error bars represent the standard deviation of 60 images. Data points are presented as mean value with error bar as SD. **d** SAXS measurement of β-lg monomer and amyloid fibrils before digestion, after gastric, and intestinal digestion. **e** STORM fluorescence images of β-lg amyloid fibrils before and after gastric digestion. The result is verified by two independent tests. Scale bars are 2 μm.

aligned parallel to the shear flow in the cylindrical capillary[38]. This allows imaging of the aligned fibrils with cross-polarized light, resulting in a bright birefringence pattern, as observed in the polarized light images in Fig. 3b. In-vitro digestion was initiated by introducing enzymatic solution to amyloid solution, and the subsequent digestion process was screened by recording microscopy images every 0.5 s. After gastric and intestinal digestion, we observed a significant decrease of the birefringence represented by the normalized intensity on the polarized light images. The final intensity was comparable to that of deionized water (Fig. 3b). Analyzing the integrated intensity of polarized light microscopy images revealed the kinetics of amyloid fibrils degradation (Fig. 3c). Specifically, upon the addition of pepsin, birefringence intensity immediately dropped, followed by an exponential decay in the first 30 min, but remained unstable with large variation in the rest of the gastric phase, which could be ascribed to the remaining short fibrils. A further decay of this signal was found in the beginning of the intestinal phase and was quickly stabilized to the range of the signal of deionized water, indicating the total absence of fibrils in the solution.

### SAXS and fluorescence imaging

The conformational diversity of β-lg monomer and amyloid fibril during different phases of digestion was investigated also by small-angle X-ray scattering (SAXS). In Fig. 3d, β-lg monomer, β-lg amyloid fibril, and deionized water showed different profiles before digestion. The β-lg amyloid fibril exhibited a higher scattering intensity in the low-q region over the monomer and water, indicating larger aggregates in the solution[39]. After gastric digestion, the profiles of β-lg monomer and digestion buffer overlapped, but showed differences from those of amyloids and deionized water. Remarkably, SAXS profiles of β-lg amyloid fibril, β-lg monomer and buffer were identical after intestinal digestion, which is confirmed by the background subtracted profiles (Supplementary Fig. 9), showing undistinguishable difference from that of deionized water. This indicates, in line with all the other experiments, that there is no detectable conformational diversity between the β-lg monomer and amyloid fibril solution after gastro-intestinal digestion. This result was verified by STORM fluorescence images (Fig. 3e), in which amyloid fibril solution exhibited a large quantity of fibrillar fluorescent structure that disappeared during gastric digestion, showing a homogenous solution with no fibrillar aggregates.

### Single-Molecule Analysis of in-vitro Digested Amyloids

While all in-vitro bulk ensemble experimental methods used above indicate full digestion of the amyloid fibrils, it certainly is possible that at the very low enzyme concentration used (trypsin activity of 0.3 U/mL i.e. >500 times lower than that required by the INFOGEST protocol[40]), some fibrils undetectable to bulk methods may not be entirely digested within the digestion time considered. The ratio of enzyme *vs.* digested protein in the in-vitro evaluation was intentionally set as low as possible to minimize autolysis/cross-digestion of the enzyme, thus enabling meaningful MALDI-MS analysis, while allowing comprehensive digestion. However, this is of no concern for two main reasons: first, even eventual residual fibrils found at these low enzymatic concentrations will be digested at larger enzyme concentrations expected in-vivo; secondly, eventual undigested remnants, will be excreted in-vivo via feces, as shown in the in-vivo sections.

We therefore performed additional experiments to clarify the minimal concentration of enzyme required to digest fibrils according to both bulk and single molecule methods. We applied AFM large-field-scan and random-scan imaging of in-vitro digested amyloids at different enzyme concentrations, from a minimum of 0.3 U/mL trypsin to progressively higher concentrations of the enzymes until no fibril fragments were detected in any samples (see Supplementary Figs. 10–11 for more information). We found that the minimum amount of trypsin enzyme required to completely digest the fibrils is 20 U/mL to 50 U/mL, which is 5 to 2 times lower than the amount of enzyme recommended for full substrate digestion by the in-vitro INFOGEST protocol. We also realized that the digestion protocol for a given enzyme concentration could be largely improved by stirring instead of shaking, requiring ca. 5–10 times less enzymes for a complete digestion (Supplementary Figs. 12–13). These results confirm both by bulk and single molecule methods that the digestion of the amyloids is completed in-vitro within the boundaries of the INFOGEST method.

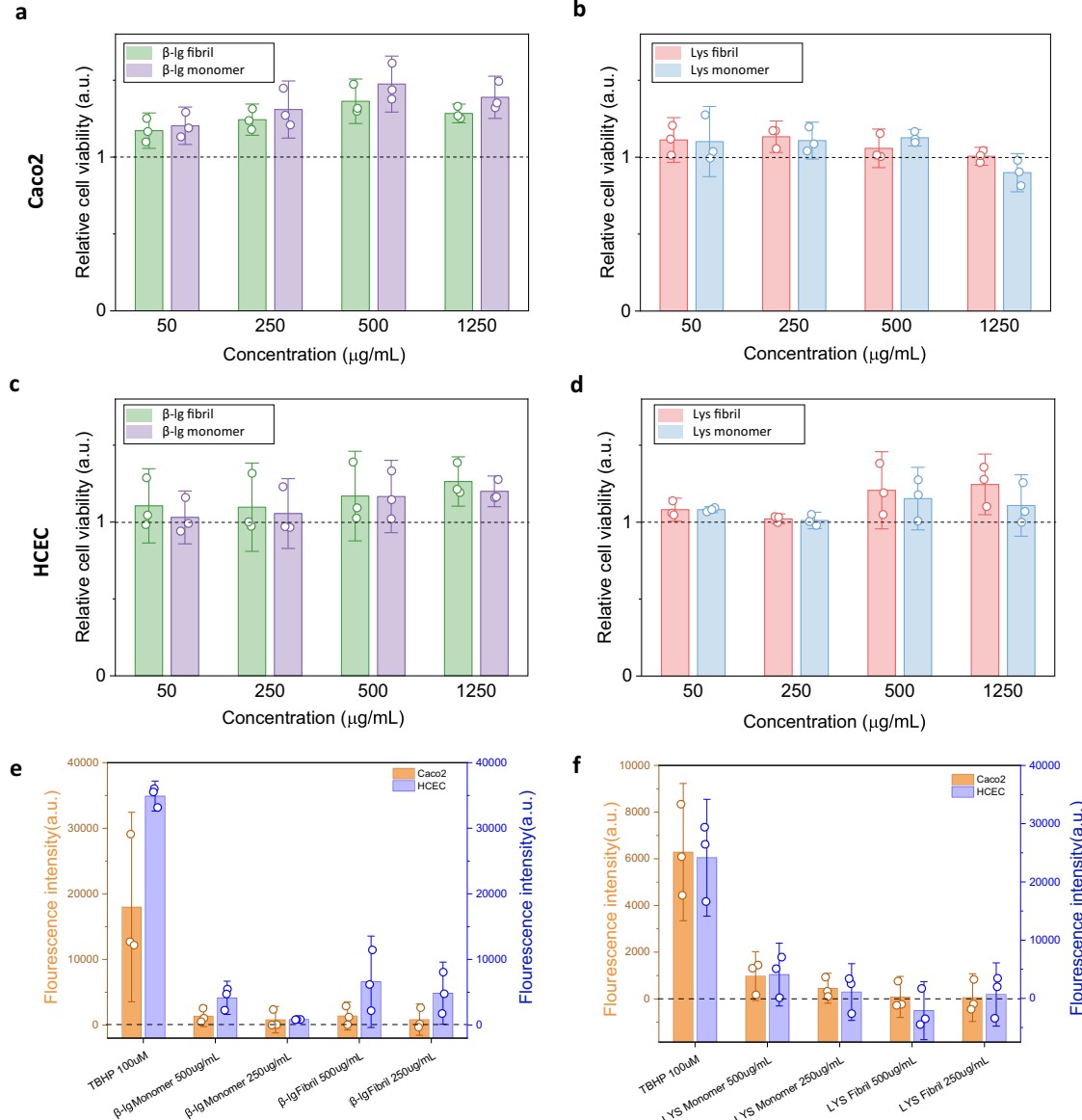

**Fig. 4 | Effect of in-vitro digested protein monomers and amyloids on cell viability.** The ratio of Caco2 cell viability treated with β-lg (**a**) and lysozyme (**b**) fibrils and monomers respectively, vs. digestion matrix at the treatment concentration of 50, 250, 500 and 1250 µg/mL. The ratio of HCEC cell viability treated with β-lg (**c**) and lysozyme (**d**) fibrils and monomers respectively, vs. digestion matrix at the treatment concentration of 50, 250, 500 and 1250 µg/mL. Reactive oxidative species experiments for (**e**) β-lg and (**f**), lysozyme digested fibrils and their monomer controls. TBHP is the positive control. The a.u. refers to arbitrary units. $N = 3$ biological replicates. The plots are showed as mean values ± standard deviation.

## Impact on viability in human cell lines

To evaluate the biocompatibility of the in-vitro digested amyloids, we first characterized their effect on the viability of two types of human intestinal cell lines (cancer coli-2, Caco2; human colon epithelial cells, HCEC). Caco2 cells are widely used to study intestinal transport, but have phenotypic characteristics of cancer cells, whereas HCEC cells are complementary in being immortalized cells with no pathological features, and therefore an attractive model for testing potential impact on healthy cells[41–43]. We characterized cellular responses of exposure to the complex buffer used for the in-vitro digestion buffers (matrix), as well as the effects of digested monomer or digested amyloid from both β-lg and lysozyme (Supplementary Fig. 15). These assays were performed for fibril and monomer concentrations ranging from 50 to 1,250 µg/mL, as well as corresponding dilutions of the digestion matrix. When we compared the viability of both cell types following exposures to digested amyloids vs. digestion matrix, or digested

monomers vs. digestion matrix (Fig. 3a), it was apparent that there was no loss of cell viability due to the presence of the digestion products. In fact, in both cell lines there was growth promotion upon exposure to digested β-lg and lysozyme fibrils or monomers relative to exposures to the digestion matrix alone (Fig. 4), consistent with an increase in bioavailable nutritive protein components.

No significant differences were found when comparing the viability of cells upon exposure to digested monomer *vs.* digested fibril, in both Caco2 and HCEC cell lines at all concentrations. The highest concentration of digestion matrix alone caused a strong reduction in cell viability (Supplementary Fig. 15), however the relative influence of the digested fibrils or monomers at any of the concentrations was not associated with any additional negative impacts (Supplementary Fig. 15). To address whether digestion is even capable to suppress toxicity of pathological amyloids, we additionally tested Aβ$_{42}$ fibrils as pathological amyloid model. Interestingly, we found that even

digested $A\beta_{42}$ fibrils did not reduce cell viability (Supplementary Fig. 16). Thus, the observations regarding overall cell viability are fully consistent with predictions from the digestion behavior, supporting that on a first-tier cytotoxicity assessment, digested fibrils are as safe as their digested monomers as there is no detectable cytotoxic effects, and furthermore, that they are well broken down to biocompatible and nutritive peptides, even in the case of $A\beta_{42}$ fibrils.

These results were further supported by reactive oxidative species (ROS) experiments, as shown in Fig. 4e, f, where for both β-lg and lysozyme fibrils and in both Caco2 and HCEC cells, fluorescence indicating the generated stress had comparable values for digested fibrils and their monomer controls.

## C. elegans model

To examine any toxic effects of feeding amyloids to *C. elegans*, we assessed the mobility of *C. elegans* (*i.e.*, swimming rates) during aging. We did not observe any detrimental effects of either monomers, fibrils, or digested fibrils on their physical activity while on the contrary, β-lg fibrils and lysozyme fibrils prolonged health span during *C. elegans* aging (Fig. 5a, Supplementary Table 2). Since the same amount of protein (0.1667 mg/ml) among monomers, fibrils, and digested fibrils were fed, we conclude that the fibrils influenced the metabolism of *C. elegans* to promote health span. We observed higher relative motilities and improved health when *C. elegans* is fed by β-lg and lysozyme fibrils (Fig. 5a, Supplementary Fig. 17, Supplementary Table 2).

Next, we asked whether fibrils could promote aggregation of endogenously expressed disease-related aggregation-prone proteins. Previously, feeding 0.1 mg/ml of human α-synuclein fibrils but not monomers to *C. elegans* promoted the endogenous aggregation of muscle-expressed human α-synuclein[39]. We chose the polyglutamine-expansion protein aggregation model that expresses human Huntington's disease polyglutamine with 35 repeats (polyQ35) tagged with YFP in the muscles of *C. elegans*, which does not aggregate on day 1 of adulthood (Fig. 5b), partially aggregated on day 5 of adulthood (Fig. 5c), and fully aggregated at day 10 of adulthood[44]. We found no change in polyQ aggregation at day 5 of adulthood when 0.25 or 1.5 mg/ml of β-lg fibrils were fed, and a mild increase of the number of polyQ aggregates in muscles when lysozyme fibrils were fed at a higher concentration (1.5 mg/ml) but not at the lower concentration of 0.25 mg/ml lysozyme fibrils (Fig. 5d, Supplementary Table 3). These results suggest only integrant, undigested lysozyme fibrils at very high concentrations may slightly affect polyQ aggregation in *C. elegans* mutants via cross-seeding, while all other cases including the digested fibrils, do not. This implies that, even for amyloid fibril cross-seeding events at extreme concentrations in polyQ *C. elegans* mutants with simple epithelia, the in-vitro human-digested amyloid fibrils are safe and trigger no cross-seeding, proving the protective mechanism of *Craniata* gastrointestinal system.

## Mouse Model

Having assessed the effect of digested amyloid fibrils on the *C. elegans* model, we turned our attention to mouse models. LC-MS/MS analysis on the fate of β-lg/lysozyme amyloids fibrils digested in-vivo by mice was applied to verify our in-vitro results. To identify the digested products of β-lg/lysozyme fibrils, contents in the intestine and colon of mice after 4 h oral administration were measured by LC-MS/MS. The heat map illustrates the identified digested oligopeptides of β-lg/lysozyme fibrils and monomers in the small intestine after digestion (Fig. 6a and Supplementary Fig. 18). A large number of low molecular weight fragments were observed due to proteolysis by intestinal enzymes, and we generally found that digested peptides of lower (Fig. 6a) or comparable molecular weights (Supplementary Fig. 18) were detected for the amyloids compared to the monomer controls, confirming in-vivo the superior digestibility of amyloids vs monomers. Most of the oligopeptide sequences from the digestion of β-lg fibrils

were also identical to those of their monomers, with the exception of LNENKV and NGECAQK, both non-amyloidogenic sequences. In the case of lysozyme digestion, all peptides observed for amyloids were also found as monomers (Supplementary Fig. 18).

Given current concerns of the possibility that amyloid cores could remain after digestion and traverse the epithelium into the blood-stream, we then sought to determine if the digested oligopeptides from β-lg/lysozyme fibrils could be detected in serum. Most importantly, after 1 h of oral administration of β-lg/lysozyme monomers and fibrils, no oligopeptides from either β-lg/lysozyme monomer nor fibrils were detected in the serum of mice (see Supplementary Table 4 for details). Unabsorbed oligopeptides proceeded to the colon, extracted and subjected to further MS analysis (Supplementary Fig. 19).

The monitoring of the distribution of β-lg fibril and monomer during digestion was performed using 5-(4,6-dichlorotriazinyl) aminofluorescein (5-DTAF) labeled-β-lg fibrils and monomer. Mice fed with labeled fibrils were sacrificed at different time points and their respective organs and blood were imaged. The migration of labeled-β-lg fibril and monomer through the intestinal tract could be clearly tracked by the high intensity regions through the 6-h time frame. Unabsorbed digests exited the intestinal tract and finally were excreted as feces (Fig. 6b, c). Further monitoring over 24 h demonstrated the absence of fluorescence in mice blood serum, and in the same vein (Fig. 6d, e), no fluorescence was also detected in the organ tissues of the mice (Supplementary Fig. 20), which corroborated no absorption of labeled peptides across epithelium into the bloodstream.

To rule out absorption of degraded fibrils through the gastrointestinal epithelia and possible targeting of other organs, we addressed the possibility of the formation in mouse tissues of amyloid plaques induced by long-term administration of β-lg/lysozyme amyloid fibrils. As a positive control, amyloid plaques were observed in Congo Red (CR)-stained brain tissue sections from the Alzheimer's disease (AD) FAD[4T] mouse group. However, no plaques were observed in the brain tissues or other main organs of mice after 30 days (Supplementary Figs. 21–22) nor after 60 days (Fig. 6f and Supplementary Fig. 23) β-lg-FL, β-lg-FH, Lys-FL and Lys-FH (where F stands for fibril, L and H stand for high and low dosage, refer to methods) administration. Thus, histological analysis appeared undistinguishable between to those from the non-AD control group fed with a normal diet and those fed dietary β-lg-F or Lys-F.

## Discussion

A putative overview of the biological fate of β-lg monomer and amyloid fibrils, after entering the gastrointestinal tract, emerging from this work is shown in Supplementary Fig. 24. Native β-lg monomers exhibiting strong pepsin resistance during gastric digestion, mainly maintain their monomeric state and partially cleave into polypeptides and oligopeptides. Undigested β-lg monomers are then broken down into oligopeptides and amino acids by intestinal enzymes. In stark contrast, β-lg amyloid fibrils are rapidly disassembled by pepsin hydrolysis, disassembling into polypeptides and oligopeptides with residual short amyloid fibrils at the end of the gastric phase. These remnants are then entirely degraded into unfolded oligopeptides and amino acids, in the following intestinal phase. The fate of these oligopeptides from there on can therefore only be postulated to be identical, that is, undergo further degradation into di-/tri-peptides or amino acids by the enzymatic activities of enterocyte membranes, before entering blood circulation either via portal vein or lymph system. A generalization of these results is inferred from additional experiments on hen egg white lysozyme amyloid fibrils digestion compared to that of native homolog monomers (found in supplementary information), where results even indicate an improved digestibility of the amyloids versus the monomers. In support of this picture, we found that β-lg/lysozyme fibrils digested in-vitro as above and then exposed to two distinct types of human epithelium cell lines, behaved essentially as their digested

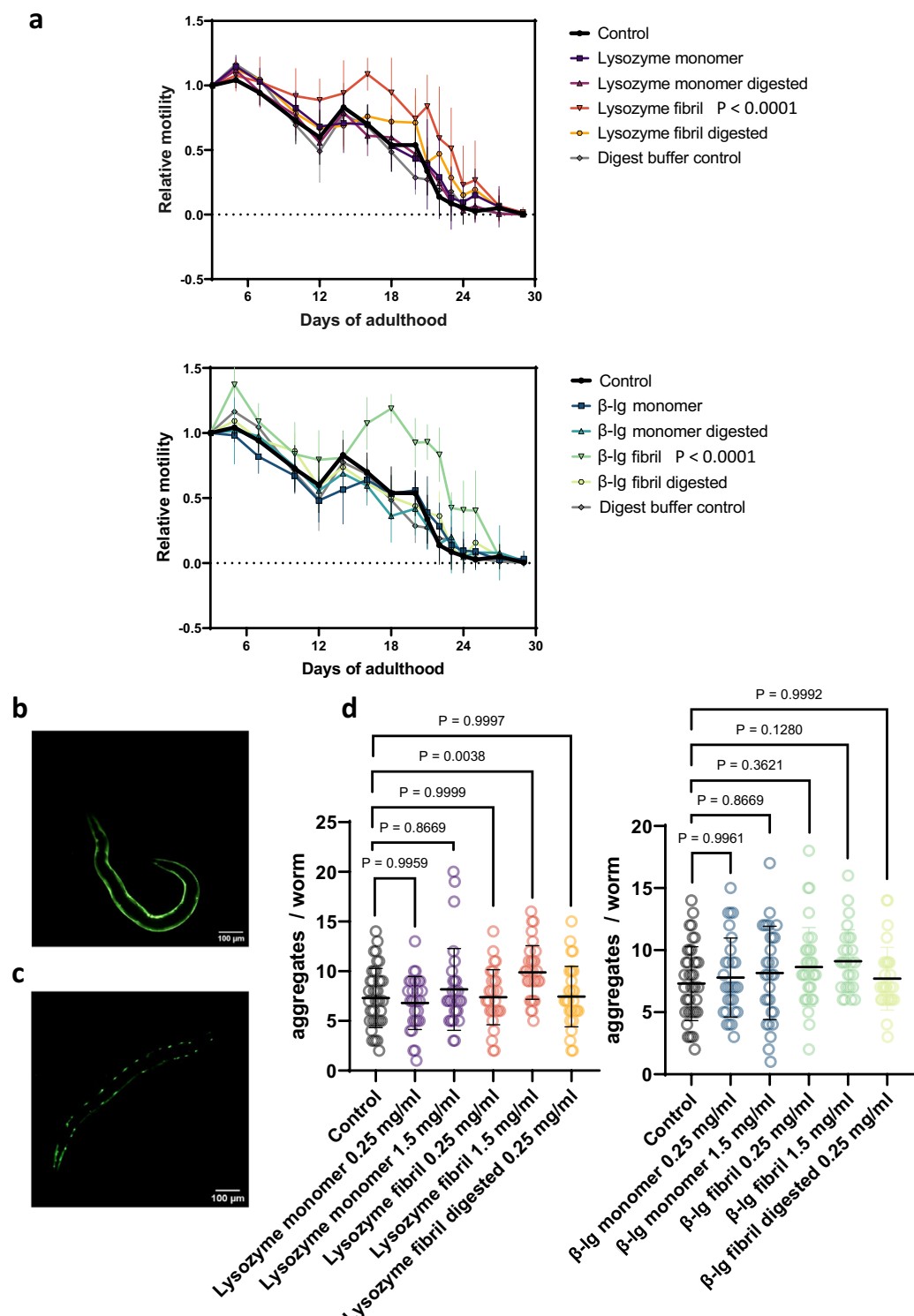

**Fig. 5 | In-vivo effects of amyloid ingestion on *C. elegans*. a** No mobility impairment in any condition after ingestion of either digested or undigested lysozyme and β-lg monomers and fibrils. By contrast, β-lg fibrils and lysozyme fibrils prolonged healthspan (active movement) of *C. elegans*. Data was analysed using a one-way ANOVA, followed by Dunnett's multiple comparisons test of the area under the curve. Proteins were supplemented at day 1 of adulthood. Data are presented as mean ± SD. The p value of the control vs. β-lg samples are β-lg monomer (0,9546), β-lg mono digested (0,9918), β-lg fibril (<0,0001) and β-lg fibril digested (0,9996). The p value of the control vs. lysozyme samples are lysozyme monomer (0,9842), lysozyme mono digested (>0,9999), lysozyme fibril (<0,0001), lysozyme digested (0,3958) and digest buffer control (0,9996). **b**, **c** Representative transgenic AM140 *C. elegans* showing of non-aggregated (day 1) (**b**) and aggregated (day 5) (**c**) Q35-YFP signal, respectively. *N* = 3 biological replicates. **d** Supplementation of protein at day 1 of adulthood leads to increased number of polyglutamine (Q35) aggregates in transgenic AM140 *C. elegans* expressing Q35-YFP in muscles ingesting a high concentration of lysozyme fibrils, but no other condition. Shown is one out of three independent biological trials. Data was analysed using a one-way ANOVA, followed by Dunnett's multiple comparisons test. Data are presented as mean ± SD.

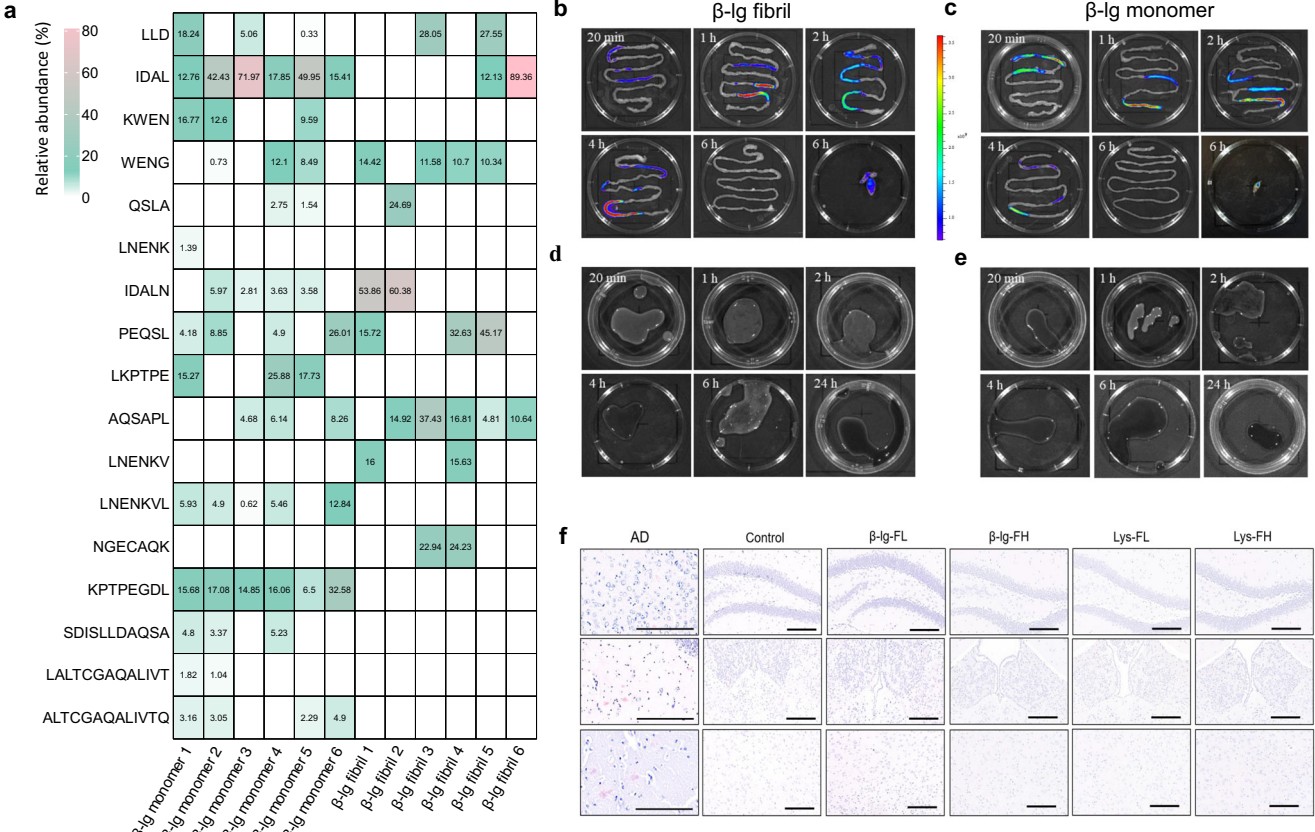

**Fig. 6 | In-vivo digestion and effects of amyloid ingestion on mice. a** Identified digested oligopeptides of β-lg monomer and fibrils in small intestine of mice after 4 h oral administration. **b, c** Fluorescence images of mice small intestines during different digestion time and feces after administration of 5-DTAF-labeled β-lg fibril (**b**) and monomer (**c**). **d, e** Fluorescence images of mice blood during different digestion time after administration of 5-DTAF-labeled β-lg fibril (**d**) and monomer (**e**). The results in panels b-e are verified by three independent experiments. **f** Representative micrographs of CR-stained mouse brain sections after 60 days of administering β-lg/lysozyme fibrils. The result was repeated with six independent experiments. Scale bars represent 200 μm.

monomeric counterparts, that is they featured no detectable cytotoxicity and produced increased viability of the cells compared to the same culture in absence of proteins, indicating that they were broken down to biocompatible and nutritive peptides.

The in-vivo results collaborate and reinforce the in-vitro evidence: no signs of cytotoxicity of digested fibrils were observed in *C. elegans* or mouse models, nor cross-seeding events from digested fibrils could be detected in the two animal models. Our mice experimental findings demonstrated that β-lg and lysozyme fibrils can be digested into the same oligopeptides as their monomers, and that any unabsorbed digested oligopeptides of fibrils do not enter the bloodstream and instead are further metabolized and excreted out of the body together with feces. Accordingly, no peptides from amyloids fibrils could be detected in the plasma of mice. In addition, no health impact concerning the induction of amyloid plaques formation by the long-term ingestion of β-lg/lysozyme fibrils were found in mice, ruling out cross-seeding mechanisms.

In summary, we applied multiple methodologies to elucidate the fate of β-lg and lysozyme amyloid fibrils first through in-vitro gastro-intestinal digestion followed by the assessment of potential in-vivo toxicity using two animal models, in both cases with β-lg and lysozyme monomers as a control. SDS-PAGE and ELISA measurements showed that β-lg amyloids were significantly degraded after gastric digestion and completely degraded following intestinal digestion. TO assay and CD spectroscopy indicated neither amyloid cores nor β-sheet conformation after gastrointestinal digestion. AFM images illustrated that amyloid fibrils could be entirely digested at enzyme:amyloid ratio within the range of the INFOGEST method specifications. MALDI-MS

spectra revealed that these digested polypeptides had molecular weights of approximately 1–2 kDa, and homolog polypeptides series were observed in the in-vitro digestion of both β-lg monomers and fibrils. Polarized light microscopy study of the kinetics of amyloid digestion showed that the birefringence signal of amyloids completely disappeared after intestinal digestion, and SAXS profiles indicated that amyloids were digested in the intestinal phase with no conformational diversity, yielding a scattering profile in each case identical to the profile of deionized water. Parallel experiments on lysozyme amyloid fibrils further reinforce these conclusions.

In-vivo results showed that no amyloid toxicity or health impact of β-lg/lysozyme fibrils degradation products were found, no crossing to the blood stream in mice and no events of cross-seeding in both polyQ35 *C. elegans* mutants and mice. Based on the above the concern of: (i) cross-seeding between remaining food protein amyloid core and pathological amyloid and (ii) cytotoxicity arising from presence of nanoparticles shall no longer hold.

This work compares comprehensively the digestibility of amyloids with their monomeric counterparts by molecular, antigenic, spectroscopic, microscopic and scattering techniques, providing the most compelling evidence that digestion of amyloids proceeds to an extent identical—if not greater—to that of the homolog monomers. The fact that food protein amyloid cannot induce toxic polypeptides after entering the human gastrointestinal tract, and further validation of these conclusions on intestinal human cell lines, *C. elegans* and mice animal models, suggest a possible classification of these ingredients—at least—as Generally Regarded as Safe (GRAS) for human consumption, although the final conclusion can only be left to the relevant

regulatory institutions. We nonetheless believe that these conclusions may open the door for promising and unexplored food and nutritional applications for food protein amyloid fibrils, introducing them as possible ingredients for human diet.

## Methods

All animal procedures complied with all relevant ethical regulations and were approved by the Laboratory Animal Committee (LAC) of South China University of Technology (AEC-2019071).

### Materials

β-lactoglobulin (β-lg, purity ≥90%) was extracted and purified from whey protein isolated according to a previously developed protocol[45], and briefly described as following. BioPURE-betalactoglobulin (lot JE 003-6-922, from 23-05-2005) from Davisco Foods International, Inc. (Le Sueur, MN) was dissolved in MilliQ water at 10 wt% concentration and adjusted to pH 4.6. Then, it was centrifuged at 15,000 rpm for 15 min at 20 °C, and the supernatant was recovered and adjusted to pH 2, before filtering through a 0.22 μm Millipore filter, followed by dialyzing first against pH 2 MilliQ water and against MilliQ water, using a Membrane (MWCO of 6 − 8k Da) at 4 °C for at least 4 h. Final solution was freeze-dried and the powder was stored at 4 °C. Lysozyme from hen egg white (L-6876), pepsin from porcine (P6887), porcine pancreatin (P7545), thioflavin T (T3516), Trichloroacetic acid (T6399), p-Toluene-sulfonyl-L-arginine methyl ester (T4626) and the rest of the reagents used were purchased from Sigma-Aldrich (Switzerland).

### Amyloid fibril preparation

β-lg and lysozyme amyloid fibrils were obtained by heating a monomer solution (2 wt%) in an oil bath at 90 °C for 6 and 24 h, respectively, with continuous stirring with a magnetic stirrer. The amyloid fibril solutions were then dialyzed (MWCO 100 kDa) against pH 2 deionized water at 4 °C for 3 days.

### Simulated gastrointestinal digestion

The INFOGEST standard protocol[40] for food digestion was applied to simulate the physiological human gastrointestinal digestion process. Various concentrations of enzymes were considered to detect the minimum required concentration of enzyme for complete digestion of the amyloids. The prepared stock solutions of simulated gastric fluid (SGF) and simulated intestinal fluid (SIF) were prepared and equilibrated to 37 °C before use. Prior to the digestion, the pepsin and trypsin activity in the pepsin and pancreatin were investigated according to the INFOGEST protocol. To begin the gastric digestion, SGF stock solution was first adjusted to pH 3 by 5 M HCl and then added to the food amyloid solution (1 mg/ml) with a ratio of 1:1 (v/v). Freshly prepared porcine pepsin solution in water was added to the mixture to a final pepsin activity of 280 U/mL in the digestion mixture. $CaCl_2(H_2O)_2$ was added to the mixed solution to achieve a final concentration of 0.15 mM prior to digestion,. The digestion procedure was carried out in a water bath (VWR 462-0493) at 37 °C with shaking condition for 120 min. An aliquot was sampled at each time point for the multiple in-vitro analysis, to which NaOH was added to inactivate the pepsin. Intestinal digestion was proceeded immediately after gastric digestion. Pancreatin was dissolved in SIF containing 0.6 mM $CaCl_2$ and added to the gastric digests with a ratio of 1:1 (v/v). The lowest used activity of pancreatin in the digestion mixture was adjusted to the low amount of protein used in our study and was 0.3 U/mg of protein. Digestion was likewise carried out for 120 min at 37 °C under shaking with periodic sampling. It should be noted that upon addition of SIF, gastric-digested lysozyme samples displayed turbidity which suggested some aggregation compared to the β-lg, possibly due to the very different isoelectric point of the two proteins. Samples were frozen to prevent further digestion at each time point before simultaneous analysis of all samples with SDS-PAGE and ELISA, while other analyses were performed immediately on fresh samples.

### Sodium dodecyl sulfate polyacrylamide gel electrophoresis (SDS-PAGE)

The dynamic proteolysis of β-lg and lysozyme monomer and amyloid fibrils during digestion were analyzed by using 4–20% stain-free precast gels (Criterion TGX Stain-Free Protein Gel, Biorad Laboratories). The digested samples were diluted with a premade Laemmli sample buffer system including 0.065 M Tris-HCl with pH 6.8, 20 mg/ml SDS, 330 mg/ml glycerol, 0.01% (w/v) bromophenol blue, after which 50 μl β-mercaptoethanol was added and heated at 90 °C for 10 min. 15 μl of diluted sample was loaded into each well, and a voltage of 200 V was applied with the running buffer solution including 0.025 M Tris-HCl, 0.192 M glycine, and 0.1% (w/v) SDS. The images were obtained by scanning the gels with a molecular imager (ChemiDoc MP Imaging System, Bio-Rad Laboratories).

### Circular dichroism (CD)

The freshly digested protein samples were diluted to 0.02 wt% using deionized water and loaded into a high-quality quartz cuvette with 1 mm optical path length. CD spectra were collected using a Jasco J-815 CD spectrometer with a wavelength ranging from 190–280 nm, bandwidth 2 nm, and scan speed 100 nm/min.

### Fluorescence assay

Thiazole orange (TO) fluorescence assay with high sensitivity and reliability was used for monitoring the presence of β-sheet from amyloid fibrils over the course of gastrointestinal digestion[46]. TO stock solution of 2 mM was prepared in dimethyl sulfoxide and stored at 4 °C before use. Gastric-digested samples were diluted with SIF to the same concentration as intestinal digested samples. 200 μl of each sample was added followed by 1 μl TO solution, and the fluorescence intensity was measured with a microplate reader (Infinite M200PRO, Tecan) using an excitation wavelength and emission wavelength of 421 nm and 455 nm, respectively. The same digestion process was likewise performed for 4 h only with digestive enzymes without monomer and fibril to determine the contribution of background fluorescence intensity. Concurrently, Thioflavin T (ThT) fluorescence was also performed to track the digestive behavior of amyloid fibrils. 2 μl of 2 mM ThT stock solution prepared in deionized water was added to each gastric-digested sample diluted with SIF. The fluorescence intensity was measured with an excitation wavelength and emission wavelength of 440 nm and 480 nm, respectively.

### Atomic force microscopy (AFM)

The digested protein solutions were collected and diluted to a final concentration of 0.01 wt.%. 10 μl of diluted sample solution was deposited on a freshly cleaved mica for 10 min, followed by rinsing with deionized water and drying by a gentle flow of compressed airflow. AFM measurements were carried out with a Bruker multimode 8 scanning probe microscope (Bruker, U.S.A.) with an acoustic hood to minimize vibrational noise. AFM imaging was operated in soft tapping mode under ambient condition with a commercial silicon nitride cantilever at a vibration frequency of 150 kHz and relatively soft tip-sample interaction. AFM images were simply flattened using Nanoscope 8.1 software.

### Determination of antigenicity

The antigenicity of all samples was evaluated using with ELISA assay using polyclonal antibodies quantification kits (AgraQuant Beta-Lactoglobulin and AgraQuant Lysozyme) purchased from Romer Labs Deutschland GmBH, Germany. All samples were diluted with deionized water to a concentration of 0.5 μg/ml for further use. To each well, 100 μl of ready-to-use standard solutions and samples were added and

incubated at room temperature for 20 min. β-lg monomer and amyloid fibrils were detected as the positive control samples, while wells with only deionized water were measured as negative controls. Wells were then emptied and rinsed five times with wash buffer, followed by 100 µl of the enzyme-conjugate solution. After 20 min of incubation followed by rinsing, 100 µl of substrate solution was added and incubated for another 20 min in the dark. The reaction was stopped by introducing 100 µl of stop solution into each microwell, and the absorbance of each sample was detected with a microplate reader (Infinite 200 PRO) at 450 nm. The milk and formula used in the control experiment were purchased from a local retail supermarket (Migros Group, Switzerland). The antigenicity of each sample was assessed according to the adsorption of a standard curve in the experiment.

### Matrix-assisted laser desorption ionization mass spectrometry (MALDI-MS)

To prepare the samples for mass determination, samples which were prior desalted using $C_{18}$ Zip Tips (Millipore, USA) were spotted onto a MALDI target (MTP 384 target polished steel TF, Bruker Daltonics, Bremen, Germany) along with 2 µl of α-cyano-4-hydroxycinnamic acid (α-CHCA) solution (10 mg/ml) in acetonitile/water/trifluoroacetic acid (50:50:0.1). We note that the singly charged protein species is not the dominant signal, and we believe this is due to the use of CHCA matrix. MALDI measurements were performed on an ultrafleXtreme MALDI-TOF/TOF mass spectrometer equipped with a smartbeam laser (Bruker Daltonics). FlexControl (Version 3.4) was used with the following programmed measurement parameters: laser frequency of 1,000 Hz in the positive-ion linear mode with acquisition ranging from 500 to 30,000 Da as well as in positive-ion reflector mode with acquisition ranging from 400 to 6,000 Da. Final spectra consisted of 1,000-3,000 shots per analysis.

### Polarized light microscopy of simulated digestion

A microfluidic system consisting of a peristaltic pump circulating the protein amyloid solution through a water bath maintained at 37 °C was set up to monitor the dynamic digestion process of amyloid fibrils through a 2 mm diameter capillary under cross-polarized light microscope (Zeiss AxioImager Z2). The gastric digestion was started by injecting pepsin into the amyloid solution with an incubation for 120 min, followed by another injection of pancreatin with an incubation for 120 min. Images were recorded using microscope software every 0.5 s. The images were subsequently analyzed using ImageJ. They were converted to 32-bit grayscale before measuring the integrated density per area, expressed as the intensity within a defined polygon in the middle of the sample jet. The intensity of a pure water jet was subtracted as background from these values.

### Fluorescence imaging

β-lg amyloid fibril before and after gastric digestion were dyed with ThT and imaged using a Nikon N-STORM microscope (Nikon, UK Ltd.) using an SR Apochromat TIRF 100 ×1.49 N. A. oil immersion objective lens. Fluorescence was detected with an EM-CCD Camera iXon DU897 (Andor). The field of view imaged typically covered 512 × 512 camera pixels corresponding to an area of ~80 × 80 µm² on the sample. An in-built focus-lock system was used to prevent axial drift of the sample during data acquisition. The laser excitation was 20% at 405 nm, with a maximum intensity measured at the tip of the optical fiber of 20 mW and an exposure time of 50 ms. The emission passed through a multi-edge dichroic filter with windows at 415–490 nm, 502–560 nm, and 660–800 nm. Images were acquired using the NIS Elements Nikon software.

### Small-angle X-ray scattering (SAXS)

SAXS experiments with Rigaku (Cu Kα radiation, $\lambda = 1.54$ Å, and a two-dimensional argon-filled Triton detector) were performed using the following conditions: the scattering vector was calibrated and centered using silver behenate, and an active range $0.014$ Å$^{-1}$ < q < $0.200$ Å$^{-1}$ was probed. Quartz capillaries with a diameter of 1.5 mm were filled with the samples and placed onto a stainless-steel holder. The samples were then measured for 2 h.

### Cell culture and cellular response assays

Caco2 cells were obtained from the Food Biotechnology group at ETH Zürich. Caco2 cells are epithelial cells isolated from colon tissue derived from a patient with colorectal cadenocarcinoma. They are cultured in DMEM (cat. Nr: 31966 021, ThermoFisher Scientifc, Massachusetts, USA) with 10% FBS (cat. Nr: 10270106, ThermoFisher Scientifc, Massachusetts, USA) 100 U/ml penicillin and 100 µg/ml streptomycin (cat. r.: 15,140,122, ThermoFisher Scientifc, Massachusetts, USA). HCEC-1CT cells were obtained from Jerry Shay from the UT Suthwestern in August 2011. HCEC-1CT cells are a colon epitelial cell line. They were cultured in ready mixed full medium (cat. Nr: MHT-039, Evercyte, Vienna, Austria). Caco2 cells were seeded at a density of 5000/cells/well in 100 µl medium in 96-well plates (cat Nr: 400096, Bioswisstec, Schhaussen, Switzerland), HCEC-1CT cells were seeded at a density of 2500 cells/well in 100 µl medium in 96-well plates (cat. Nr: 734-0079, Corning, New York, USA). ATP was quantified, as an indicator of metabolically active cells (cat. Nr:G7571, Promega, Dübendorf, Schweiz). Caco2 and HCEC cells were exposed to the digested Lysozyme and ß-lg monomers and fibrils at defined concentrations for 4 h. The digested fibrils/monomers samples were freeze dried and reconstituted in deionized water. After INFOGEST digestion, the digested protein samples or digestion matrix were subject first to heat-shock (95-degree, 10 min) to inactivate the enzymes and were then freeze-dried. Afterwards the samples were diluted in cell culture media to reach final concentrations. A portion (100 µl) of each sample was added per well to the cells. Triton X-100 (0.01 %) was used as a positive control. At the end of the 4 h treatment, media containing the digested substances was removed and replaced with 45 ul of fresh media. Afterwards 45 ul of CellTiter-Glo reagent was added to each well. The plate was shacked 2 min on an orbital shaker and incubated 10 min in the dark. Afterwards 75 ul were transferred to a white half area 96 well plate. Luminescence with 1000 ms integration time was measured using a plate reader (Infinite M200 Pro, Tecan group, Männerdorf, Switzerland).

### C. elegans Strains

TJ1060 spe-9(hc88) I; rrf-3(b26) II. AM140 rmIs132 [unc-54p::Q35::YFP]

### Mobility assay

Mobility assay was adapted from Statzer et al.,[47]. In brief, gravid TJ1060 *C. elegans* adults were synchronized by bleaching and L1 arrested over night. L1 worms at day 1 of adulthood were then transferred into U-shaped 96-well plates in S-complete buffer and heat-inactivated bacteria. At day 1 of adulthood the protein samples or controls were added to the wells for a final concentration of 0.1667 mg/mL. From day 3 of adulthood onwards the *C. elegans* were scored for mobility using the MicroTracker (MTK100). For statistical analysis, the area under the curve was measured, and the mean for each run was calculated. Statistical analysis was performed by using a 1-way ANOVA.

### Q35 aggregate quantification

Poly-Q35:YFP aggregates were imaged with an upright bright field fluorescence microscope (Tritech Research, model: BX-51-F) using the triple band filter sets consist of the 69000 ET-DAPI/FITC/TRITC (69000x, 69000 m, 69000bs, EX/EM 25 mm, ringed, DC 25.5 × 36 x 1 mm) (Chroma Technology, catalog number: 69000)[48]. Transgenic AM140 *C. elegans* were fed with protein samples from day 1 of adulthood onwards and imaged at a time-point when aggregation started but the worms were not yet fully aggregated, usually day 4 of

adulthood. Images were then analyzed using ImageJ. An intensity threshold was set manually based on the control worms and used for all conditions. Then aggregates were counted using the 'Analyze particle' tool. To minimize noise, a minimum value for circularity (0.02–0.03) and size (2–3 pixel) was set.

## Mouse strains
Kunming mice were obtained from Beijing Vital River Laboratory Animal Technology Co., Ltd. Male mice (5 weeks old) were housed in enclosures on sawdust bedding at 25 °C and 40% relative humidity including a normal light cycle (12 h dark /12 h light cycle). Deionized water and fodder were provided. Mice were acclimatized for one week before in-vivo experiments. All animal procedures complied with all relevant ethical regulations and were approved by the Laboratory Animal Committee (LAC) of South China University of Technology (AEC-2019071).

## LC-MS/MS analysis on In-vivo mice gastrointestinal digests
The in-vivo digested products of β-lg/lysozyme amyloid fibrils and monomer and their peptides identification in small intestinal tracts, colon and serum were studied by LC-MS/MS analysis[49]. Six-week-old Kunming male mice ($n = 6$ per group) were starved for 24 h, after which they were orally gavaged with 200 µl 1 mg/ml of β-lg/lysozyme amyloid fibrils and monomer solutions, separately. The contents of each small intestinal tracts and colon were collected after 4 h gastrointestinal digestion for digests identification. For serum β-peptides identification, mice were orally gavaged with 200 µl 1 mg/ml β-lg/lysozyme amyloid fibrils and monomer solution, serum samples were collected after 1 h digestion.

LC-MS/MS measurements were performed on a Q Exactive™ mass spectrometer coupled to Easy nLC (Thermo Fisher Scientific). Specifically, 15 µL of each sample was injected from the autosampler to Zorbax 300SB-C18 peptide traps (Agilent Technologies, Wilmington, DE), and then separated by liquid chromatography column (15 cm long, 75 µm I.D., 5 µm resin, C18-reversed phase column) which was equilibrated by buffer A (0.1% formic acid solution) and then separated with a linear gradient of buffer B (0.1% formic acid in 84% acetonitrile) at a flow rate of 250 nL/min controlled by IntelliFlow technology over 120 min. MS data was acquired using a data-dependent top10 method which dynamically chose the most abundant precursor ions from the survey scan (300–1800 m/z) for high-energy collisional dissociation (HCD) fragmentation. The target value was determined based on predictive automatic gain control (AGC) and the dynamic exclusion duration was 20 s. Survey scans were acquired at a resolution of 70,000 at m/z 200 and resolution for HCD spectra was set to 17,500 at m/z 200. Survey scans were acquired at a resolution of 70,000 at m/z 200 and HCD spectra resolution was set to 17,500. Normalized collision energy of 27 eV was used while the underfill ratio, which specifies the minimum percentage of target value likely to be reached at maximum fill time, was set at 0.1%. peptide recognition mode was enabled throughout the run. MS data were processed using the MaxQuant software (version 1.5.5.1) by UniProt database without specifying enzyme cleavage rules for the processed MGF files. The following search parameters were used: ±20 ppm for peptide mass tolerance, 0.1 Da for MS/MS tolerance. Variable modification: Oxidation (M). Label-free peptide quantification (LFQ) based on extracted ion chromatograms and validation were performed in the MaxQuant software. The cutoff value of global false discovery rate (FDR) for peptide identification was set to 0.01.

## In-vivo distribution of fibrils during digestion
Six-week-old Kunming male mice ($n = 3$ per group) were orally gavaged with 200 µl 1 mg/ml β-lg monomer and amyloid fibrils solution, respectively, which labeled using the dye 5-(4,6-dichlorotriazinyl) aminofluorescein (5-DTAF). Small intestine, blood, brain, heart, liver, spleen, lung, kidney and feces of mice from each time points were collected, and then imaged using an in-vivo imaging system (IVIS, Lumina XRMS III, PerkinElmer).

## Long-term intake of fibrils
Six-week-old Kunming male mice were randomly separated into five groups ($n = 8$ per group) namely with the control group, β-lg-FL group, lys-FL group, β-lg-FH group and lys-FH group (F stands for fibril; L for low dosage; H for high dosage). The control group had no further treatment, the β-lg-FL group was daily orally gavaged with 200 µL β-lg-fibrils of 1 mg/ml, the lys-FL group with 200 µl lysozyme fibrils of 1 mg/ml, β-lg-FH group of 5 mg/ml and lys-FH group of 5 mg/ml. The mice were sacrificed after 30 and 60 days, respectively, of administration and sections of the mainly tissues were processed for amyloidosis examination. The major organs including brain, heart, liver, spleen, kidneys and lungs were excised and fixed in 10% formalin. Congo red (CR) staining method was used for identification of amyloids in tissue sections, brain sections of Alzheimer's disease (AD) mice were used as positive control.

## Reporting summary
Further information on research design is available in the Nature Portfolio Reporting Summary linked to this article.

## Data availability
The data used to reproduce the results and to support the findings are available within the article, Supplementary Information file and the Source Data file. The Source Data are provided with this paper [https://doi.org/10.6084/m9.figshare.24201405]. The mass spectrometry proteomics data have been deposited to the ProteomeXchange Consortium via the PRIDE partner repository with the dataset identifier PXD045698 [https://www.ebi.ac.uk/pride/archive/projects/PXD045698]. Extra data are available from the corresponding authors upon request. Source data are provided with this paper.

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

## Acknowledgements

The authors gratefully acknowledge Dr. Serge Chesnov from the Functional Genomics Center Zurich and Pinotsi Dorothea from the Scientific Center for Optical and Electron Microscopy (ScopeM), ETH Zurich. D.X. also acknowledges the Guangzhou Elite Project for financial support (GEP No: JY202032). Chia-Wei Lin is kindly acknowledged for assistance with the MALDI analysis. R.M. wishes to posthumously acknowledge Sir. Christopher Dobson for inspiring past discussions on the digestion of amyloid fibrils.

## Author contributions

R.M. conceptualized the project; R.M., J.Z. and S.J.S. supervised the research and designed studies; D.X, J.Z., W.L.S., I.K., S.D., A.M. and S.H. conducted the experiments. D.X., J.Z., W.L.S., I.K., A.M., S.H. B.L., L.L.,

C.Y.E. and M.R. contributed to experimental and data analyses. J.Z., D.X. and R.M. wrote the manuscript and all authors contributed to evaluating data and writing the final version.

## Competing interests

The authors declare no competing interests.
