## [Peer Review File · Nature Communications]

REVIEWER COMMENTS

Reviewer #1 (Remarks to the Author):

Xu et. al. set out to probe if amyloid fibrils of beta-lactoglobulin (β -lg) and lysozyme can be substituted as superior food protein compared to their monomeric counterpart. The authors use a wide range of biophysical techniques such as AFM, TEM, MALDI-MS, CD, and SAXS to assess the degraded products after gastrointestinal digestion of these food amyloids in vitro. The authors demonstrated no detectable conformational diversity among the monomer and amyloid fibrils of β -lg after gastrointestinal digestion. The study further suggests that most of the oligopeptide sequences formed from the digestion of β -lg fibrils were identical to that of their monomers, which were analyzed by an array of in vitro and in situ methods. The study was further extended in in vivo model organisms such as *C. elegans* and mice to consolidate the possibilities of food amyloid fibrils as safe and potential ingredients for human nutrition. However, they found no detectable toxicity or evidence of cross-seeding to pathological amyloids upon long-term administration in mice.

Overall, the manuscript reflects a good amount of work done by the authors. However, the conclusions drawn from the study are not completely convincing. Therefore, my opinion is that it suffers from major conceptual and experimental flaws, as explained below:

1. Firstly, I speculate that the amyloid fibrils formed by β -lg and lysozyme are not from full-length monomeric protein but rather from some fragmented products. The authors should run SDS-PAGE or perform MALDI to verify the molecular weight of the amyloid fibrils formed, keeping the respective monomeric protein as a control. In this light, the MALDI-MS data shown by the authors in Fig. 3b (β -lg undigested fibrils) and S14b (undigested lysozyme fibrils) clearly shows that the amyloid fibril formed is not from the full-length protein as there is no peak at the respective molecular weight of β -lg (18 kDa) and lysozyme (14 kDa) in the spectra. This clearly showing that the amyloid fibrils the authors are using are not formed from the full-length protein but rather from fragmented products, as evidenced by multiple lower molecular weight peaks in the MALDI-MS spectra. Therefore, all the following experiments carried out by the authors are questionable.
2. When the author commented on the allergen formation by β -lg monomer, they should clearly mention which full-length segment they refer to and whether this segment is further digested during amyloid fibril digestion. The authors claim that the allergenic epitope of β -lg is inside the protein core in a monomeric state, whereas this is exposed on the surface of amyloid fibrils. Does any previous study prove this observation?
3. "In essence, this shows that the previously undigestible β -lg monomer became much more digestible after fibrilization starting from the gastric track" How is it possible? In the amyloid field, it is well established that amyloid fibrils are more resistant to pH and protease digestion than their monomeric counterparts. The authors need to explain the mechanism of protease action where monomeric protein is resistant and fibrils are prone to degradation.

4. It is skeptical to believe the degradation profile shown in Fig 1c. I suggest that the authors use 15% Tris-tricine gel, which is recommended for resolving low molecular weight proteins, rather than a gradient SDS-PAGE, which is typically used when the sample has proteins with a wide range of molecular weight. This will enable the authors to resolve smaller digested fragments further. Also, Fig 1c further validates point 1, that the undigested fibrils of β -lg are not formed from the full-length protein.

5. "Digestion in the intestinal phase, however, proceeded slowly due to the aggregation of gastric-digested amyloid fibrils upon the addition of intestinal fluid, possibly resulting in the retardation of digestion by pancreatin." This is not clear at all. What does the author mean by aggregation of gastric-digested amyloid fibrils? Does the smaller fibril fragment aggregate, or the peptide further aggregate? Justify this statement.

6. Rather than doing the TO fluorescence, I suggest that the author do mass spectrometry along with the HPLC. This data will reveal the protein digestion profile; otherwise, nothing is evident from the digestion profile.

7. Authors have used the bulk technique CD spectroscopy to detect the presence of β -sheet content before and after gastrointestinal digestion. This is acceptable for analyzing the monomeric protein structure. Authors further claim that the food amyloids are decomposed into random coil polypeptides after gastrointestinal enzymatic reaction, with no detectable β -sheet residues in the protein solution from CD spectra. CD spectroscopy accounts for the bulk measurement of the protein solution. But the author should centrifuge the sample after digestion to examine the amyloid fibrils and then perform CD spectroscopy to affirm the results. Otherwise, smaller amounts of fibrils might not reveal their presence in CD spectra due to the ensemble measurement.

8. It would be interesting to verify if pathological amyloids such as A β -42 fibrils will exhibit similar or different observations as β -lg and lysozyme. The authors should use A β -42 fibrils as a control for all the experiments. If the author wants, they can use amyloid fibril by folded protein such as prion as their control. I fail to see why suddenly amyloid of β -lg and lysozyme should show different behavior in contrast to other amyloid fibrils.

9. The author must explain why there are "larger globular aggregates in the size range of tens of nanometers after gastric digestion, and forming multiple nano-islands after intestinal digestion". What does this data imply? Are these the aggregated forms of small digested peptides? If so, are they toxic? The authors should characterize the cytotoxic profile to validate this.

10. In vitro studies involving gastric digestion and intestinal digestion of β -lg and lysozyme amyloid fibrils show similar profiles in CD. However, they exhibit completely different profiles in AFM. The authors need to address this discrepancy in observation. They need to perform centrifugation at each step to determine how much digestion leads to soluble peptide fragments and amyloid fibrils or aggregates.

11. The authors must also include a control where the simulated digestive/intestinal fluid is used at similar pH without the digestive enzymes. This will help them to have better insight into the biological fate of the food amyloids.

12. The authors have performed MALDI TOF, but I would suggest this type of study requires LC-MS data to convince the reader about the digestion profile in parallel with centrifugation, showing the percentage of pelletable mass versus soluble peptide.

13. The SAXS data is of inferior quality and it is difficult to deduce whether there is a significant difference between the samples. The author claims monomeric β -lg is less digestible than β -lg amyloid fibrils in simulated gastric fluid. However, the SAXS data for gastric-digested samples reveal no significant difference between the β -lg monomer and gastric buffer (per Author's claim). While in contrast, the β -lg amyloid fibril exhibits higher scattering intensity. This contradicts the author's observation that amyloid fibrils are more prone to gastric digestion leading to fragmentation than the monomeric protein.

14. In the in vivo experiments, the authors must use the amyloid fibrils and monomer of A β -42 as a control to determine toxicity.

15. When the β -lg and lysozyme fibrils were fed to *C. elegans*, the authors should use the labeled fibrils to assess the fate of the fibrils. Also, they should verify whether any co-fibrils are formed by β -lg or lysozyme fibrils with polyQ.

16. The mouse experiments are an interesting approach. The author should use the amyloid fibrils of A β -42 as a positive control of toxicity. These experiments and data are very preliminary.

17. The aggregation of neurodegenerative proteins can initiate in the gut and easily transmit to the brain via the gut-brain axis and propagate the disease pathology. In this scenario, it could be possible that these food amyloid supplements, even in ultra-low quantities, may cross-seed the aggregation of pathogenic heterologous amyloid proteins, which underlie various neurodegenerative disorders. A recent report also suggests heterologous cross-seeding of lysozyme with other IDPs such as α -Syn has been reported (PMID: 33539920). Similarly, previous studies also established that the fragmented amyloids formed from infectious bacteria in the gut can cross-seed A β peptide and initiate the A β amyloidosis (PMID: 32999841). However, the authors claim that these β -lg food amyloids are getting fragmented by gastro-intestinal digestion, there is a high chance of heterologous cross-seeding with other amyloid proteins in the gut by surface-mediated secondary nucleation and may propagate the brain pathology through the gut-brain axis. Therefore, one cannot neglect previously established findings and widely-accepted facts about the cross-seeding phenomenon of amyloid seeds and its pathological implications. In this study, the authors used fluorescently labeled fibrils to assess the distribution. However, the fragment products will not be detectable using the fluorescence technique. For detecting pico or femto molar seed concentration, they need to perform radiolabelled assays to determine the presence and localization of seed conclusively.

18. For the CR-stained tissue experiment, the load of the amyloid seed might not be enough to manifest pathological symptoms (plaques) in the given experimental period (30 days). This does not imply the absence of the cross-seeding phenomenon; rather, the load/duration might not be sufficient for plaque formation.

19. No information regarding the ethical clearance for the in vivo study has been mentioned in the methodology section.

20. The axis is missing in certain figures, such as Fig. 3 and S14.

Overall, I disagree with the author's assessment that amyloid fibrils can be considered a safe food for human consumption. A much more systematic and long-term toxicity study needs to be done to disprove the safety concerns. Also, even a minute amount of fibril seeds goes to the brain or circulates in the blood; it is impossible to detect them until the load is high enough to manifest physical symptoms. I cannot recommend the publication of this manuscript, much less so in Nature communications.

Reviewer #2 (Remarks to the Author):

In their manuscript entitled "Food Amyloid Fibrils as Safe Nutrition Ingredients: In-vitro and In-vivo Assessment", Xu, Mezzenga et al. explore the safety aspects of using food protein amyloid fibrils as potential ingredients for human nutrition. The authors use a vast portfolio of methods, including in vivo approaches in animal models, to address the central question as to whether potentially pathologic amyloidogenic core structures remain after passage of fibrillar protein aggregates through the digestive system. This is obviously an important question in the field of food chemistry as it has to be ruled out conclusively that remaining amyloid nuclei can exert nanoparticle-related cytotoxicity or even induce pathological amyloid structures via cross-seeding, before protein amyloids can be considered as food additives. The authors show that monomeric and fibrillar forms of selected food proteins are digested into essentially the same, non-toxic oligopeptides and consider both as equally safe, paving the way for protein fibrils as beneficial food ingredients.

Although the major message is convincingly supported by the experiments overall, the manuscript appears a bit like compiled in a rush. There are some shortcomings with the description of the methods used and the presentation of the data obtained, in particular related to the mass spectrometric experiments, which need to be addressed to improve the quality of the manuscript. Specifically, the mass spectra in Fig. 3 need scaled y-axes, ideally showing absolute intensity in the same range per spectra set to allow for semi-quantitative comparison. While appropriate for the average mass signals in the protein spectra from linear mode, the monoisotopic mass signals in the peptide spectra from reflector mode should be annotated by masses with two digits in line with the high resolution/mass accuracy of the latter mode. As signals can in principle also originate from autolysis/cross-digestion of the enzymes involved (or from CHCA matrix clusters at least in the low m/z range of the spectrum), the data would be much more convincing if the signals would be mapped onto the respective substrate protein sequence in the sense of a peptide mass fingerprint. This can be done via a table comparing calculated and observed masses in the supplementary information or, more elegantly, by a database search of the peak lists (similar to what is done later with the LC-MS data). Along the same lines, the signal-to-sequence assignment should be corroborated by mass spectrometric sequencing of the respective peptides, at least for species explicitly mentioned in the text like the dominant peaks common/differential in Ig monomer and fibril, respectively. Given the instrument used for generating the data shown, MALDI-TOF/TOF-MS/MS is accessible to the authors.

In general, it is recommended that this part of the text is proofread by an expert in mass spectrometry, as there are several issues that need correction/commenting:

- The calculation of the multiply charged species is incorrect as the mass of the proton is neglected; for example, the doubly charged species is $(18366+2)/2$
- It is unusual that the doubly and not the singly charged species is the dominant signal in a MALDI mass spectrum (see Fig. 3a). This is likely due to the use of CHCA as matrix, which is the most commonly used matrix for peptides, but not for proteins. The authors may want to consider to comment on that phenomenon, which may otherwise confuse readers expecting a 'normal' charge distribution
- Double-check terminology: mass spectroscopy vs. mass spectrometry (correct form); reflection mode vs. reflector mode (correct form)

While the necessary information on MALDI-MS is provided in the Methods section, the description of the proteomic analysis of the in vivo digestion products is absolutely insufficient, particularly in the absence of any references for the methods used. First, details are needed for the C18 SPE of the intestinal/serum samples (loading/washing/elution). For serum samples, was the cartridge eluted in a way that peptides were separated from serum proteins? Or was it a bulk elution of all peptides/proteins bound to the stationary phase? The latter case may explain why no peptides were detected in the serum samples. As to the LC-MS conditions, the minimal set of information should include: LC system, column(s) used, mobile phases, gradient, mass spectrometer type, data acquisition mode, resolution/AGC targets, mass range. Similarly, information on data processing should include search parameters (no enzyme setting? Cys-status?) and quantification method (LFQ?). A complete protein/peptide list extracted from the MaxQuant results must be part of the supplementary information (an incomplete table like the one shown as Table S3 does not tell anything).

Another severe shortcoming in data presentation is the statistics. In the text related to Fig. 2 c-f, the authors claim to couple AFM imaging with "comprehensive statistical analysis". However, no details on the statistical test(s) applied can be found in the Methods section, nor are any p-values reported in the legend/displayed in the figure panels. Figure S8 is a particular bad example in this regard, as it even does not display error bars and is thus meaningless.

Minor issues (referring to which is hampered as no page and line numbers are provided):

- General: as the headlines of the Results section only contain the methods applied (mostly as abbreviations), the manuscript reads a bit like an old-fashioned article from the field of analytical sciences and does not come with the more modern flow the broad-interest readers are used to in Nature Communications. The authors may want to consider to use headlines expressing the major findings and to provide a short summary/conclusion at the end of each section. Along similar lines, the

Discussion mainly summarizes the data and is rather weak on what is gained for the field of food chemistry by the findings presented. Valid for all parts of the text, the authors tend to use very long sentences (see for example the last sentence of the section 'AFM and TEM imaging') and should consider rephrasing into shorter, separate sentences.

- The allergenicity plots in Fig. 1d,e and Fig. S2 are hard to read for the non-expert and may need some more explanation, particularly as it does not become clear from the Methods section which ELISA assays were used. For example, the authors explicitly refer to the high initial allergenicity of Ig-monomers in Fig. 1d, but the initial allergenicity of the Ig fibril seems to be about twice as high(?). Here, readers from outside food chemistry/allergology may need more guidance. In general, it would be helpful to keep consistent the figure design of Fig. 1b,c/ Fig. 1d,e and their corresponding supplementary Figures Fig. S1/S2, e.g. red arrows pointing to protein bands, color shading for milk etc. (not explained in the legend).
- TO and ThT fluorescence assays are shown for Ig fibrils, but only TO for lysozyme fibrils. Is there a reason for not showing ThT for the latter?
- What is the size of the scale bars in Fig. 2a? Likely not 100 nm as given in the legend to panel b.
- Results in vivo, end of first paragraph: the authors conclude that the fibrils influenced the metabolism of *C. elegans*. Isn't it the improved digestibility of the fibrils that is meant?
- The distribution of monomers/fibrils is monitored after fluorescent labeling with an amine-reactive dye. This means that labeled lysine residues escape cleavage by trypsin, which is likely the major protease in pancreatin. This should be discussed.
- SDS-Page vs. SDS-PAGE (correct form)
- AD mice: the mouse model used is not mentioned
- Methods section: "... were analyzed using the proteomics method." There is no one proteomics method
- Fig. S9: this reviewer is not an expert in AFM, but negative heights on the color scale may need explanation.
- Milli-Q water is a brand name, may be replaced by deionized water

Reviewer #3 (Remarks to the Author):

The manuscript by Xu et al. examined the safety of two main types of food amyloids, i.e., whey protein beta lactoglobulin (blg) and plasma protein lysozyme. The central strategy of the study was to mimic the digestion of food amyloids in the gastric and intestinal tracts with the introduction of two enzymes in pepsin and pancreatin, and to further characterise the products post the enzymatic activities using a range of analytical techniques of AFM, absorbance, SDS-PAGE, CD, fluorescence, MALDI-MS, polarised

light microscopy as well as SAXS. In all instances, the products were shown convincingly to be low in containing a cross-beta backbone characteristic to amyloid fibrils and were down to the level of peptides post digestion. The biocompatibility and hence safety of the amyloid fibrils was validated by behaviour assays employing *C. elegans* and mice in vivo models.

The analytical (including MALDI-MS) and animal behavior data presented in this study were of top quality, as expected from this leading group in functional amyloid structure and food protein science. As a prime goal of this study was to demonstrate the non-toxic nature of food amyloids, the authors are suggested to complement their analytical-behavior study with additional toxicity data (e.g., ROS, apoptosis, and immune response using cell culture or the animal tissue) to fully establish the whole safety parameters for food amyloids. By including such cellular/tissue toxicity data this study would be more complete in the reviewer's opinion and could serve as a comprehensive benchmark study for guiding future biological and environmental applications of functional amyloids.

Other suggestions/minor issues:

1. Fig. 5 and its relevant text, here the abbreviation "pre-digested" referred to already digested amyloids, in contrast to the other sample of "amyloids", but the term sounded like for amyloids prior to digestion. Consider modifying the abbreviation to avoid confusion.
2. Fig. S20 was an overview of the biological fate of β -lg monomer and amyloid fibrils after entering the gastrointestinal tract. This scheme might be better placed in the main text for readability.
3. Minor language issues, such as in Abstract: "...monomers from (a) technological, nutritional, sensorial, and physical perspective(s)"; in Introduction: " Homo sapiens has (have) been consuming dairy..." "...may reach the bloodstream and nucleating (nucleate) the formation of pathological amyloids by cross-seeding..." "it was therefore chosen as (a) main model in this digestion study"; in Results: "Similarities of the peptide sequences on β -lg amyloid fibrils after fibrillization with some of the IgE-binding epitopes from earlier studies highly supports (support) that"...

**INDIVIDUAL REVIEWER COMMENTS**

Reviewer #1 (Remarks to the Author):

Xu et. al. set out to probe if amyloid fibrils of beta-lactoglobulin (β -lg) and lysozyme can be substituted
as superior food protein compared to their monomeric counterpart. The authors use a wide range of
biophysical techniques such as AFM, TEM, MALDI-MS, CD, and SAXS to assess the degraded
products after gastrointestinal digestion of these food amyloids in vitro. The authors demonstrated no
detectable conformational diversity among the monomer and amyloid fibrils of β -lg after gastrointestinal
digestion. The study further suggests that most of the oligopeptide sequences formed from the digestion
of β -lg fibrils were identical to that of their monomers, which were analyzed by an array of in vitro and in
situ methods. The study was further extended in in vivo model organisms such as C. elegans and mice
to consolidate the possibilities of food amyloid fibrils as safe and potential ingredients for human
nutrition. However, they found no detectable toxicity or evidence of cross-seeding to pathological
amyloids upon long-term administration in mice.

Overall, the manuscript reflects a good amount of work done by the authors. However, the conclusions
drawn from the study are not completely convincing. Therefore, my opinion is that it suffers from major
conceptual and experimental flaws, as explained below:

1. Firstly, I speculate that the amyloid fibrils formed by β -lg and lysozyme are not from full-length
monomeric protein but rather from some fragmented products. The authors should run SDS-PAGE or
perform MALDI to verify the molecular weight of the amyloid fibrils formed, keeping the respective
monomeric protein as a control. In this light, the MALDI-MS data shown by the authors in Fig. 2b (β -lg
undigested fibrils) and S14b (undigested lysozyme fibrils) clearly shows that the amyloid fibril formed is
not from the full-length protein as there is no peak at the respective molecular weight of β -lg (18 kDa)
and lysozyme (14 kDa) in the spectra. This clearly showing that the amyloid fibrils the authors are using
are not formed from the full-length protein but rather from fragmented products, as evidenced by
multiple lower molecular weight peaks in the MALDI-MS spectra. Therefore, all the following
experiments carried out by the authors are questionable.

Admittedly, we are both surprised and puzzled by this comment by the Reviewer. We are of course well
aware that β -lg and lysozyme amyloids-as those described in this work-are formed by hydrolysed (and
not full sequence) peptides, and this by at least one decade: the Reviewer can refer to our own early
(2011) work: <https://pubs.acs.org/doi/abs/10.1021/bm200216u> and all the publications which have
followed since then. There is not a single point in the manuscript where we have stated or suggested the
contrary.

Therefore, we can reassure the Reviewer on this point that is not necessary to "speculate" (as per
his/her own words) on this, also because this is clearly shown in Figure 2 in our submission with the
mass distribution of the initial undigested fibrils. Similarly, the statement "Therefore, all the following
experiments carried out by the authors are questionable" is difficult to understand: in fact, we clearly
show by a multitude of techniques both in-vitro and in-vivo **that hydrolysis progresses throughout the**
**GI tract** (from initially partly hydrolysed segments in the case of amyloids), **via progressively smaller**
**peptides**, so that this opening statement appears more a misunderstanding than a real concern.

We therefore hope we have clarified this point/misunderstanding based on our own earlier work.

2. When the author commented on the allergen formation by β -lg monomer, they should clearly mention
which full-length segment they refer to and whether this segment is further digested during amyloid fibril
digestion. The authors claim that the allergenic epitope of β -lg is inside the protein core in a monomeric
state, whereas this is exposed on the surface of amyloid fibrils. Does any previous study prove this
observation?

While we may not be able to identify the full-length segment of the protein which is detected by the
ELISA allergenicity test (as any larger segments containing the minimum allergenic epitope would also
give a positive result), it is clear that "this segment is further digested during amyloid fibril digestion" as
the Reviewer can note by looking at Figure 1e, where the ELISA signal decrease with digestion time.
We have removed the sentence on the positioning of the allergenic epitope to avoid any ambiguity:

"~~highly supports that these allergenic motifs are presented on the surface of amyloid fibrils after~~
~~fibrillization, indeed corroborating that these allergenic epitopes are more susceptible to pepsin~~
~~hydrolysis in the amyloid form"~~

3. "In essence, this shows that the previously undigestible β -lg monomer became much more digestible
after fibrilization starting from the gastric track" How is it possible? In the amyloid field, it is well
established that amyloid fibrils are more resistant to pH and protease digestion than their monomeric
counterparts. The authors need to explain the mechanism of protease action where monomeric protein
is resistant and fibrils are prone to degradation.

Concerning the point that amyloid fibrils are well established to be more resistant than monomers, we
must respectfully disagree with the Reviewer. Contrary to the pathological amyloid assembled by the full
length proteins, the food amyloid used in this paper is fabricated by heating 6-15 hours at pH 2. While
low pH is used in the present work to prepare the initial food amyloid fibrils via protein hydrolysis and
then self-assembling, there are no studies that demonstrate a higher resistance of food amyloid fibrils to
stomach and intestine proteases compared to their monomer counterparts, quite the contrary: the
Reviewer can refer for example to the works of Singh et al (<https://doi.org/10.1021/jf101722t>). If, on the
contrary, the Reviewer is aware of any such study, it should point us to the relevant work. Most certainly,
the Reviewer refers to other types of amyloids (pathological, bacterial) exposed to different type of
proteases -not digestion-, and not within the GI tract, which is far from the conditions and experiments
presented in this work.

In any event, food amyloids are digested into smaller peptides than the monomer counterparts as
demonstrated in the present study by the SDS-Page, Fluorescence, CD, MALDI (in-vitro) and LS-MS
experiments (in-vivo) carried out and presented in the manuscript. There is no ground to object to these
conclusions given the consistency among all the techniques used.

To summarize: β -lg monomer is a notoriously difficult protein to digest as largely documented in
literature; amyloid fibrils being partly hydrolyzed from the beginning do favour proteolysis and further
degradation.

4. It is skeptical to believe the degradation profile shown in Fig 1c. I suggest that the authors use 15%
Tris-tricine gel, which is recommended for resolving low molecular weight proteins, rather than a
gradient SDS-PAGE, which is typically used when the sample has proteins with a wide range of
molecular weight. This will enable the authors to resolve smaller digested fragments further. Also, Fig 1c
further validates point 1, that the undigested fibrils of β -lg are not formed from the full-length protein.

We understand the point raised by the Reviewer and we have taken actions to rebut his/her concerns.
The gradient SDS-PAGE was run when sample has proteins with a wide range of molecular weight. In
fact, during monomer/fibril digestion, we expected a large degradation of monomer/fibrils into a full
spectrum of molecular weight, especially lower than 10 kDa. We specifically planned to have a gradient
gel to better visualize this broad molecular range, including the presence of enzymes. More details
within 10 kDa on the SDS-PAGE can be seen in the MALDI result.

However, we agree with the point made by the Reviewer and we have further run a new SDS-PAGE gel
as suggested, with higher molecular resolution at lower molecular weights to resolve the digested
products better than what is shown in Fig 1c, i.e. the lowest band is set at 2 kDa. The gel is now
presented as supplementary **Figure S1**: as the reviewer can see, the hydrolysis is unambiguous, which
we hope will clear his/her residual concerns. We have also added the following sentence in the
manuscript: "**An SDS-PAGE gel with a higher resolution at lower molecular weights is shown in Fig. S1**"

Fig. S1. SDS-PAGE gel (15%) of β -Ig amyloids showing a high resolution at the low molecular weight
components during intestinal digestion, demonstrating full hydrolysis after complete gastrointestinal
digestion.

Regarding to Reviewer's argument that the β -Ig fibril are not from full-length protein, again please refer
to the response in point 1 and 3. Besides our work we want to point out that the process of full-length β -
Ig protein hydrolysis into hydrolyzed polypeptides before/during fibrillization is well understood and can
be seen by SDS-PAGE also in the literature produced by other groups (see for example
<https://pubs.rsc.org/en/content/articlelanding/2015/SM/C5SM01037C>); this has been also observed on
other food proteins such as ovalbumin (<https://pubs.acs.org/doi/full/10.1021/bm301481v>), oat-globulin
(<https://onlinelibrary.wiley.com/doi/full/10.1002/advs.202104445>), etc.

5. "Digestion in the intestinal phase, however, proceeded slowly due to the aggregation of gastric-
digested amyloid fibrils upon the addition of intestinal fluid, possibly resulting in the retardation of
digestion by pancreatin." This is not clear at all. What does the author mean by aggregation of gastric-
digested amyloid fibrils? Does the smaller fibril fragment aggregate, or the peptide further aggregate?
Justify this statement.

We agree that the sentence was not clear. What we meant is that upon gastric->intestine pH change,
the linear charge density of peptide must change and cross the isoelectric point, and thus induce
aggregation. We have rephrased the sentence for more clarity.

Following the suggestion, "Digestion in the intestinal phase, however, proceeded slowly due to the
aggregation of gastric-digested amyloid fibrils upon the addition of intestinal fluid, possibly resulting in
the retardation of digestion by pancreatin" has been modified as: "A slower digestion in the intestinal
phase might be due to the change of the degraded polypeptides charge upon the gastric to intestinal pH
transition with possible aggregation of these polypeptides" in the revised manuscript.

6. Rather than doing the TO fluorescence, I suggest that the author do mass spectrometry along with
the HPLC. This data will reveal the protein digestion profile; otherwise, nothing is evident from the
digestion profile.

We believe this is another misunderstanding. TO was only performed to stain beta sheets of amyloid
nuclei/fibrils: neither mass spectroscopy nor HPLC detect integrant nuclei or amyloid fragments.
However, in the manuscript we have already performed both MALDI TOF in vitro and LC-MS in vivo,
which are both much more sensitive than HPLC. The conclusion of LC-MS, in particular, is that no
residual fragment in the intestinal tract from β -Ig and lysozyme amyloids is amyloidogenic. Again, mass
spectroscopy results suggested by the Reviewer are already present in the manuscript and consistent
with TO.

7. Authors have used the bulk technique CD spectroscopy to detect the presence of β -sheet content
before and after gastrointestinal digestion. This is acceptable for analyzing the monomeric protein
structure. Authors further claim that the food amyloids are decomposed into random coil polypeptides
after gastrointestinal enzymatic reaction, with no detectable β -sheet residues in the protein solution from
CD spectra. CD spectroscopy accounts for the bulk measurement of the protein solution. But the author
should centrifuge the sample after digestion to examine the amyloid fibrils and then perform CD
spectroscopy to affirm the results. Otherwise, smaller amounts of fibrils might not reveal their presence
in CD spectra due to the ensemble measurement.

We are most thankful to the Reviewer for this comment and for suggesting additional experiments to
clarify this further. Indeed, while all bulk experimental methods (ensemble methods as the Reviewer
correctly call them) used for the in-vitro part do indicate full digestion of the amyloid fibrils, it is certainly
possible that *at the very low enzyme concentration used (trypsin activity 0.3 U/mL, more than 300 times
lower than INFOGEST protocol required)*, some fibrils -undetectable to bulk methods- may not be
entirely digested within the digestion time considered. We must note that the ratio of enzyme vs
digested protein in the in-vitro part was intentionally chosen as low as possible precisely to limit to a
minimum the autolysis/cross-digestion of the enzyme to allow meaningful MALDI-TOF analysis, while
allowing complete digestion (see also our reply to Reviewer 2). However, this is of no concern for two
main reasons: first, even eventual residual fibrils found at these low enzymatic concentrations will be
digested at larger enzyme concentrations expected in-vivo; secondly, eventual undigested remnants will
be excreted in vivo via feces as we show in the in vivo part.

Nonetheless, inspired by the comment of the Reviewer, we have run additional experiments to clarify
the minimal concentration of enzyme required to digest fibrils according **to both bulk and single**
**molecule methods**. We therefore applied AFM random imaging and confirmed the result by AFM large
field imaging (100*100 μm^2) on in-vitro digested amyloids at different enzyme concentrations starting
from the minimal concentration (those for the *in-vitro* part) and moved to progressively larger

concentration of the enzymes until no fibril fragments were detected at any location of the samples. We
do prefer this technique to the one suggested by the Reviewer, because centrifugation cannot separate
eventual amyloid nuclei from the enzymes, due to comparable molecular weights. After all, even after
such a process, CD remains still an ensemble, bulk method.

By doing so, we find that the minimum amount of enzyme required to digest the fibrils in full is 50 U/mL
for trypsin activity in the shaking condition; and around 10 U/mL in the stirring condition. We note,
however, that this amount is still much less than half of the amount of enzyme recommended by the in-
vitro INFOGEST protocol (ref. 40 in the manuscript) to digest entirety of the substrate, therefore, **the**
**entire digestion of the amyloids** is now demonstrated in vitro within the boundaries of the INFOGEST
method by both bulk and single molecule methods. We are thankful to the Reviewer for having
suggested these experiments which strengthen further the in-vitro part.

The following text and images were added:

**Single-Molecule Analysis of in-vitro Digested Amyloids**

While all in-vitro bulk ensemble experimental methods used above indicate full digestion of the amyloid
fibrils, it certainly is possible that at the very low enzyme concentration used (trypsin activity of 0.3 U/mL
i.e. >500 times lower than that required by the INFOGEST protocol⁴⁰), some fibrils undetectable to bulk
methods may not be entirely digested within the digestion time considered. The ratio of enzyme vs.
digested protein in the in-vitro evaluation was intentionally set as low as possible to minimize
autolysis/cross-digestion of the enzyme, thus enabling meaningful MALDI-TOF analysis, while allowing
comprehensive digestion. However, this is of no concern for two main reasons: first, even eventual
residual fibrils found at these low enzymatic concentrations will be digested at larger enzyme
concentrations expected in-vivo; secondly, eventual undigested remnants, will be excreted in-vivo via
feces, as shown in the in-vivo sections.

We therefore performed additional experiments to clarify the minimal concentration of enzyme required
to digest fibrils according to both bulk and single molecule methods. We applied AFM large-field-scan
and random-scan imaging of in-vitro digested amyloids at different enzyme concentrations, from a
minimum of 0.3 U/mL trypsin to progressively higher concentrations of the enzymes until no fibril
fragments were detected in any samples (see Figs. S10-S11 for more information). We found that the
minimum amount of trypsin enzyme required to completely digest the fibrils is 20 U/mL to 50 U/mL,
which is 5 to 2 times lower than the amount of enzyme recommended for full substrate digestion by the
in-vitro INFOGEST protocol. We also realized that the digestion protocol for a given enzyme
concentration could be largely improved by stirring instead of shaking, requiring ca. 5-10 times less
enzymes for a complete digestion (Figs. S12-S13). These results confirm both by bulk and single
molecule methods that the digestion of the amyloids is completed in-vitro within the boundaries of the
INFOGEST method.

**Figure S10.** AFM images of β -Ig amyloid fibrils after gastrointestinal digestion in the INFOGEST protocol
under the shaking condition, at different concentrations of trypsin ranging from 0.3 U/mL to 100 U/mL.
The fibrils are fully digested with trypsin concentration of around 50 U/mL. AFM images were collected
in random locations on the mica surface at a size of 15 by 15 μ m.

Confirmation of full digestion of amyloid fibril after *in-vitro* gastrointestinal digestion in the shaking condition at the trypsin activity of 50 U/mL and 100 U/mL by AFM large field scanning of 100*100 μm².

	Total length of fibril per area (nm/μm ²)			average
No digestion	2015.20	1395.10	2442.46	1950.92±526.63
Shaking-0.3U/ml	58.38448	137.65216	204.30097	133.45±400.34
Shaking-2U/ml	116.17881	10.13063	21.94973	49.42±58.12
Shaking-20U/ml	0	0	113.21422	37.73807
Shaking-50U/ml	0	0	19.72806	6.57602
Shaking-100U/ml	0	0	0	0

Figure S11. AFM statistical analysis of β-Ig amyloid fibril length after gastrointestinal digestion in the INFOGEST protocol under the shaking condition. The total length of fibril per area was calculated at different concentrations of trypsin ranging from 0.3 U/mL to 100 U/mL.

Figure S12. AFM images of β-Ig amyloid fibrils after gastrointestinal digestion in the INFOGEST protocol under the moderate stirring condition, at different concentrations of trypsin ranging from 0.3 U/mL to 100 U/mL. The fibrils are fully digested with the trypsin concentration of around 20 U/mL. AFM images were collected in random locations on the mica surface at a size of 15 by 15 μm.

Confirmation of full digestion of amyloid fibril after *in-vitro* gastrointestinal digestion in the stirring condition at the trypsin activity of 50 U/mL and 100 U/mL by AFM large field scanning of 100*100 μm².

	Total length of fibril per area (nm/μm ²)			average
No digestion	2015.20	1395.10	2442.46	1950.92±526.63
Shaking-0.3U/ml	15.90687	28.70346	28.43687	24.35±7.31
Shaking-2U/ml	5.15418	10.66382	46.8535	20.89±22.65
Shaking-20U/ml	0	0	0	0
Shaking-50U/ml	0	0	0	0
Shaking-100U/ml	0	0	0	0

Figure S13. AFM statistical analysis of β-Ig amyloid fibrils after gastrointestinal digestion in the stirring INFOGEST protocol. The total length of fibril per area were calculated at different concentrations of trypsin ranging from 0.3 U/mL to 100 U/mL.

8. It would be interesting to verify if pathological amyloids such as Aβ-42 fibrils will exhibit similar or different observations as β-Ig and lysozyme. The authors should use Aβ-42 fibrils as a control for all the experiments. If the author wants, they can use amyloid fibril by folded protein such as prion as their control. I fail to see why suddenly amyloid of β-Ig and lysozyme should show different behavior in We contrast to other amyloid fibrils.

As the title of the manuscript states very clearly, this work is about **food** amyloid fibrils fate through the gastrointestinal tract. Aβ-42 and even less prion (responsible for the only infective amyloid disease) proteins are not food grade, but toxic peptides, which furthermore do not naturally occur *in vivo* within the gastrointestinal tract. As such, studying pathological amyloids digestion is neither within the scope of the paper, nor a natural administration pathway to be explored *in-vivo*. Instead, we have used blg and lysozyme monomers as food-grade controls and benchmarked any conclusions against these two food proteins, for which safety is already known and well established.

Nevertheless, inspired by the reviewer, and to explore the digestion effects in the pathological realm, we did perform the investigation of Aβ-42 fibril digestion and studied the effect of *in-vitro* digested amyloids on cell viability (Fig. S16). We found that Aβ-42 fibril can also be mostly digested at the trypsin activity of 0.3 U/mL, and the digested fibril showed no detectable impact on the cell viability on both Caco2 and HCEC cells, compared to their digestion matrix at all concentrations. While food amyloid digestion is the focus of this manuscript, these results show that even pathological fibrils would become mostly non-toxic to the GI tract when digested. These experiments actually follow up from a discussion the corresponding author (Mezzenga) and Sir Prof. Chris Dobson, had back in 2017 at IDP 2017, Mohali, India, and are full in line with the expectation back then. This is why Prof. Dobson is now duly mentioned in the acknowledgments.

 Fig. S16. The digestion of Aβ42 fibrils and the effect of in-vitro digested protein amyloids on cell viability.
 (a) The digestion of Aβ42 fibrils upon the trypsin enzyme activity of 0.3U/mL. The Aβ42 fibrils were
 obtained at the concentration of 50 μM in the buffer of 1:1 mixture of 0.1% NH4OH and 100 mM Tris
 buffer (with 0.02% NaN3 and pH 7.4). (b) the relative Caco2 and HCEC cell viability in the left panel
 treated with digested fibril vs. digestion buffer at the treatment concentration of 6, 30, 60 and 150 μg/mL.
 Cell viability was evaluated on the basis of quantification of ATP present as an indicator of metabolically
 active cells, performed using the CellTiter-Glo luminescent cell viability assay. Cells were exposed for
 4 h. Each cell viability measurement was performed in triplicate and repeated three times. The cell
 viability values for exposed cells were normalized to corresponding values for the same cells incubated
 in the same cell culture medium without addition of Aβ42 and enzymes. Values shown are mean
 values ± SD.

 9. The author must explain why there are “larger globular aggregates in the size range of tens of
 nanometers after gastric digestion, and forming multiple nano-islands after intestinal digestion”. What
 does this data imply? Are these the aggregated forms of small digested peptides? If so, are they toxic?
 The authors should characterize the cytotoxic profile to validate this.

The Reviewer is here referring to the text part commenting (former) Figure 2a: the digested monomers.
 Digested monomers from blg and lysozyme are known to be non-toxic and there is no need to
 characterize cytotoxicity of such digested food protein monomers. Rather, the nano-island observed by
 AFM both from digested monomers and amyloids are the result of solvent evaporation and substrate
 deposition as already explained in the text a few lines below and as conclusively demonstrated via an
 additional filtration experiment:

*“The observed nano-islands in both digested monomer and fibrils are believed to be the assemblies of*
 *polypeptides derived from monomer and fibril digestion due to drying and surface absorption of the mica*
 *substrate. To confirm this, we filtered the intestinal digested β-Ig monomer and amyloid solution with 10*
 *kDa membrane filters with a pore size less than 2 nm. The AFM images of filtrate also show nano-*
 *islands for both samples (Fig. S6) with similar heights of approximately 1 nm, although in lower quantity,*
 *which is attributable to their loss in the filtration process. These results were confirmed by transmission*
 *electronic microscopy (TEM) in Fig. 2b and S7.”*

However, the new AFM section has been expanded, modified and strengthened according to the other
 comments of this same Reviewer (see point 7), so that while we have just replied to the Reviewer
 concerns, this part of the text is no longer in the manuscript.

10. In vitro studies involving gastric digestion and intestinal digestion of β-Ig and lysozyme amyloid fibrils
 show similar profiles in CD. However, they exhibit completely different profiles in AFM. The authors

need to address this discrepancy in observation. They need to perform centrifugation at each step to
determine how much digestion leads to soluble peptide fragments and amyloid fibrils or aggregates.

We thank the reviewer for pointing out that digested β -Ig and lysozyme fibrils exhibit different profiles on
AFM although they both showed random coil in CD. The primary reason is the sequence of β -Ig and
lysozyme and their charge on their polypeptides after in-vitro digestion. The full-length β -Ig and
lysozyme have very different isoelectric points, 5.2 and 11 respectively, which means their short
polypeptides are overall differently charged while preparing AFM sample on mica surface. As mica
surface is slightly negatively charged, these polypeptides assemble and sediment on the surface (we
proved the nano-islands are the assemblies of the polypeptides by the filtering tests in Fig. S6).
Another reason is the hydrophobicity of the digested β -Ig and lysozyme polypeptides after in-vitro
digestion. We also expect some differences while depositing these polypeptides on the hydrophobic
mica surface.

11. The authors must also include a control where the simulated digestive/intestinal fluid is used at
similar pH without the digestive enzymes. This will help them to have better insight into the biological
fate of the food amyloids.

This is an excellent suggestion, as this allows further demonstrating that the digestion down to
oligopeptides is due mostly to enzymes and not pH. We have performed these experiments and revised
the manuscript accordingly.

Fig. S14. AFM image of β -Ig amyloid fibrils after gastrointestinal digestion without digestive enzymes.
The fibrils remained intact overall but tended to attach together which is believed due to the enlarged
the ionic strength by mixing with the SGF and SIF buffer.

12. The authors have performed MALDI TOF, but I would suggest this type of study requires LC-MS
data to convince the reader about the digestion profile in parallel with centrifugation, showing the
percentage of pelletable mass versus soluble peptide.

Again, we believe this is a misunderstanding: all the mouse in-vivo peptide analysis is based indeed on
LC-MS. MALDI-TOF is only one of the several methods used for the in-vitro introductory part of the
manuscript. The LC-MS is already carried out extensively in the most prominent part of the manuscript
(in vivo GI tract and in vivo serum).

13. The SAXS data is of inferior quality and it is difficult to deduce whether there is a significant
difference between the samples. The author claims monomeric β -Ig is less digestible than β -Ig amyloid
fibrils in simulated gastric fluid. However, the SAXS data for gastric-digested samples reveal no
significant difference between the β -Ig monomer and gastric buffer (per Author's claim). While in
contrast, the β -Ig amyloid fibril exhibits higher scattering intensity. This contradicts the author's
observation that amyloid fibrils are more prone to gastric digestion leading to fragmentation than the
monomeric protein.

SAXS is used to detect the eventual presence of aggregates such as fibrils and/or amyloid nuclei. Once
the peptides are dissolved down to the molecular level (soluble peptides), SAXS is no longer capable of
detecting differences. Therefore, SAXS provides no information with respect to the molecular distribution
of soluble peptides.

The signal shown in Figure 3d central panel, shows (as the Reviewer noted) higher signal of gastric-
digested blg amyloid compared to gastric-digested blg monomer, i.e. residual presence of aggregates,
which is totally consistent with the former figure 2a central panel, and with our AFM analysis in general.
This rather demonstrates, that SAXS is indeed of accurate enough sensitivity to measure presence of

residual aggregates, at least at the same level of all the other bulk techniques used for the in-vitro
characterization. Importantly, after the in-vitro digestion by the intestine (Figure 3d, rightmost panel), all
the curves are overlapping, i.e there are no more aggregates/fibrils/nuclei detectable in suspension.
These results are therefore not at all in contradiction with the observation that amyloid fibrils are more
prone to gastric digestion than the monomeric protein but are in full agreement with the other bulk
methods used in the *in-vitro* part.

14. In the in vivo experiments, the authors must use the amyloid fibrils and monomer of A β -42 as a
control to determine toxicity.

Again, this manuscript is about **food amyloid fibrils**; controls must be of food grade nature (i.e. blg and
lysozyme monomers). Nonetheless, as requested by the Reviewer, we have investigated A β 1-42
digestion and applied the digested remnants to two epithelium cells lines, showing that even
pathological amyloids, when digested, feature no noticeable cytotoxicity. Please refer to our response to
point 8. In-vivo digestion of A β 1-42 to mice and *C. elegans* is out of the scope of this manuscript.

15. When the β -lg and lysozyme fibrils were fed to *C. elegans*, the authors should use the labeled fibrils
to assess the fate of the fibrils. Also, they should verify whether any co-fibrils are formed by β -lg or
lysozyme fibrils with polyQ.

We agree with the reviewer that following the fate of these fibrils would be a great idea. To address this,
we performed the following experiment: monomers, fibrils, and digested fibrils were stained with TO and
fed to *C. elegans* for 24 h. Shown below are the confocal images of the pharynx and head regions. As
TO emits green fluorescence upon binding to amyloid fibrils, we indeed observed green fluorescence in
the pharynx, intestine, and hypodermis. While these observations indicated and agreed well that the
proteins/peptides got ingested and digested by *C. elegans*, and finally some staining in the hypodermis,
suggests this got spread across tissues. We realized that TO also bound non-specifically to bacterial
aggregated protein and endogenous *C. elegans* aggregated proteins. This is also supported by the
lowering of endogenous age-related protein aggregation and lifespan extension of *C. elegans* with the
feeding aggregating-staining dyes (DOI: 10.1038/nature09873). Thus, given that aggregating-staining
dyes will stain endogenous *C. elegans* aggregates and that these dyes influence physiology and protein
homeostasis, the experiment suggested by the reviewer is unfortunately not technically feasible.

The polyQ experiments do indicate that there is an enhancement of age-related PolyQ aggregation only
when very high levels of integrant lysozyme fibrils were fed to *C. elegans*. This was not observed when
lower levels were used and most importantly, not when exposed to digested lysozyme fibrils.

16. The mouse experiments are an interesting approach. The author should use the amyloid fibrils of A β -42 as a positive control of toxicity. These experiments and data are very preliminary.

Again, within the scope of food amyloid digestion in mouse study, pathological amyloids such as A β -42 is not a meaningful control for the reasons already commented at point 8 and 14. Controls must be of food grade nature, hence we have used the monomers from which the food amyloids are produced. In-vivo digestion of A β 1-42 to mice and *C. elegans* is out of the scope of this manuscript.

The experiments on mice are performed (among other methods) by LC-MS, which is the most accurate method to provide mass distribution and identification of peptide sequences.

17. The aggregation of neurodegenerative proteins can initiate in the gut and easily transmit to the brain via the gut-brain axis and propagate the disease pathology. In this scenario, it could be possible that these food amyloid supplements, even in ultra-low quantities, may cross-seed the aggregation of pathogenic heterologous amyloid proteins, which underlie various neurodegenerative disorders. A recent report also suggests heterologous cross-seeding of lysozyme with other IDPs such as α -Syn has been reported (PMID: 33539920). Similarly, previous studies also established that the fragmented amyloids formed from infectious bacteria in the gut can cross-seed A β peptide and initiate the A β amyloidosis (PMID: 32999841). However, the authors claim that these β -lg food amyloids are getting fragmented by gastro-intestinal digestion, there is a high chance of heterologous cross-seeding with other amyloid proteins in the gut by surface-mediated secondary nucleation and may propagate the brain pathology through the gut-brain axis. Therefore, one cannot neglect previously established findings and widely-accepted facts about the cross-seeding phenomenon of amyloid seeds and its pathological implications. In this study, the authors used fluorescently labeled fibrils to assess the distribution. However, the fragment products will not be detectable using the fluorescence technique. For detecting pico or femto molar seed concentration, they need to perform radiolabelled assays to determine the presence and localization of seed conclusively.

On this point we must respectfully disagree with the Reviewer for several reasons.

To start, the gut-brain axis hypothesis (in the context of amyloidosis) relies on disturbances along the
brain-gut-microbiota axis, via inflammation of either the central nervous system and/or the enteric
nervous system which is then associated to a decreased gut-blood barrier, leading to increased
permeation of potential pathogens into the blood. A possible parallel and simultaneous mechanism
which has been hypothesized would be the eventual cross-over of microbiota amyloids into the blood
stream, with possible cross-seeding to pathological amyloid proteins. However, as we show in the
present work, the peptides originating from digested amyloids are equal or smaller in sequence than
those obtained from the digestion of the corresponding monomers: because the monomers of blg and
lysozyme are known to be food-grade and non-toxic (hence they do not produce inflammation), the
digested homolog fibrils must be at least equally safe, since they are digested down to even smaller
peptide sequences than the monomers. **Accordingly, no inflammation can be triggered by the
amyloids over the monomers and this is a known result:** in 2017 we tested the whole blood
glutathione (GSH) concentration in mice fed with β -Ig amyloids and we found no changes (Shen et al.
Nature Nanotechnology 2017, 12, 642 - DOI: 10.1038/NNANO.2017.58). Similarly, we do not detect
crossing into the bloodstream of any amyloidogenic sequences from the digested proteins which could
eventually be capable of cross-seed pathological amyloids. In short, the gut-brain axis is not a possible
mechanism in the present case.

Secondly, the Reviewer mentions two studies which support cross-seeding of pathological amyloids
from fragmented fibrils.

The first study suggested is by Claessens et al (PMID: 33539920): this works studies cross-seeding of
alpha synuclein from β -Ig and lysozyme fragmented amyloid fibrils, exactly as in the present case. The
study concludes that when using fragmented (not digested!) amyloid fibrils, lysozyme but not β -Ig can
cross-seed alpha synuclein amyloids: these conclusions are identical to these we achieve in polyQ
mutant *C. elegans*, that is only integrant undigested lysozyme but not β -Ig amyloids may cross seeds
polyQ amyloids. Therefore, this rather demonstrate, that *C. elegans* is a good model to test cross-
seeding and hence it reinforces the value of our conclusions **where no cross-seeding is observable in
C. elegans by fully digested lysozyme and β -Ig amyloids.**

The second study (PMID: 32999841) is co-authored by the corresponding author of the present
submission (Mezzenga): Once more, the work studies cross-seeding of $A\beta$ initiated by fragmented (not
digested) bacterial amyloids, which again does not contradict in any way the absence of cross seeding
observed in our work, where digested fibrils are studied instead.

In summary, none of the works contradict our findings, but rather they are supportive of our findings (at
least the first work); the statement "*Therefore, one cannot neglect previously established findings and
widely-accepted facts about the cross-seeding phenomenon of amyloid seeds and its pathological
implications*" is therefore incorrect because it refers to literature using only fragmented and not digested
fibrils.

About the suggestion of using radiolabeled isotopes: we do have at ETH Zurich an isotope facility within
the very same institute of the corresponding author of this submission which we have already used in
the past (Shen et al. Nature Nanotechnology 2017): contrarily to what was stated by the Reviewer,
radiolabeling and isotopes detection methods are only sensitive to the presence of individual atomic
isotopes and cannot resolve peptide sequences, hence, not applicable to reveal amyloidogenic nuclei;
the technique used in vivo in the present work, LC-MS, is the most sensitive technique capable of
detecting and of resolving peptide sequences with a sensitivity of 0.1 ng per mL, i.e. 0.1 ppb. Given the
high feeding dosage in our mice studies, this technique is sufficiently sensitive to detect the smallest
presence of amyloidogenic nuclei, allowing to exclude the cross-seeding hypothesis in mice precisely as
we independently concluded from the study with *C. elegans*.

A final comment: the Reviewer argues that eventual seed concentration should be detected down to the
pico (10^{-12}) or even femto (10^{-15}) molar concentration to rule out potential presence of seeds;
however this would be by far below the critical aggregation concentration (CAC) of amyloid seeds which
is (for $A\beta$) of the order of 90 nM, that is 10^{-8} molar (<https://www.nature.com/articles/s41598-018-19961-3>). In other words, at concentrations below 10^{-8} molar (10^4 to 10^7 larger than what
mentioned by the Reviewer), any amyloid seed would already dissolve, i.e. no cross-seeding can occur
at these concentrations because no seeds can exist below CAC. We close by noting that the CAC of 90
nM for $A\beta$ corresponds to 3.6 10^{-4} g/L, a thousand time higher than the sensitivity of the LC-MS
technique we have used to rule out the presence of amyloidogenic seeds in the bloodstream. For β -Ig
amyloids, the CAC is even reported larger (<https://link.springer.com/article/10.1007/s11483-009-9101-3>),
so that no reasonable possibility to miss the presence of amyloid seeds crossing the gut barrier can be
expected in the present work based on the techniques used.

18. For the CR-stained tissue experiment, the load of the amyloid seed might not be enough to manifest
pathological symptoms (plaques) in the given experimental period (30 days). This does not imply the
absence of the cross-seeding phenomenon; rather, the load/duration might not be sufficient for plaque
formation.

We thank the Reviewer for this important comment. Let us start by noting that the AD mice (C57BL/6) used have an age of 6 weeks, yet plaques are fully detectable by CR; the feeding of amyloids to mice described in the earlier version of this work lasted for 30 days, i.e. a realistically long time span to detect the possible presence of plaques, since nine mouse days are equivalent to one human year (<https://doi.org/10.1016/j.lfs.2015.10.025>), and thus 30 days is not a short period of time.

However, we fully agree with the Reviewer that a longer feeding time would make the conclusions stronger and indeed, we have run a longer amyloid feeding timeframe for mice: we have now extended the feeding to 60 days, that is 100% longer compared to the experiment presented in the original submission and equivalent to nearly 7 years human continuous feeding of amyloids. The results confirm and reinforce the absence of pathological plaques by cross-seeding, and this not only in brain, but also the other major mice organs (heart, spleen, kidney, lung, liver). We thank the reviewer for having suggested this important experiment strengthening our conclusions.

**Fig. 6f. Representative micrographs of CR-stained mice brain sections after 60 d of administering β -Ig/lysozyme fibrils. Scale bars represent 200 μ m.**

**Figure S23. Micrographs of CR-stained major organ tissue sections from different mice groups after 60 d of feeding food amyloid (where F stands for fibril, H and L stand for high and low dosage). Scale bars represent 200 μ m.**

19. No information regarding the ethical clearance for the in vivo study has been mentioned in the
methodology section.

The Reviewer has possibly received a previous version of the manuscript. The one provided to the
Editor has a section disclosing the ethical approval:

**"Mouse strains: Kunming mice were obtained from Beijing Vital River Laboratory Animal Technology Co.,**
**Ltd. Male mice (5 weeks old) were housed in enclosures on sawdust bedding at 25 °C and 40% relative**

humidity. Deionized water and fodder were provided. Mice were acclimatized for one week before in
vivo experiments. All animal procedures complied with all relevant ethical regulations and were
approved by the Laboratory Animal Committee (LAC) of South China University of Technology
(AEC-2019071).”

20. The axis is missing in certain figures, such as Fig. 3 and S14.

We have fixed this error in the revised manuscript. Thank you for noting this.

Overall, I disagree with the author's assessment that amyloid fibrils can be considered a safe food for
human consumption. A much more systematic and long-term toxicity study needs to be done to disprove
the safety concerns. Also, even a minute amount of fibril seeds goes to the brain or circulates in the
blood; it is impossible to detect them until the load is high enough to manifest physical symptoms. I
cannot recommend the publication of this manuscript, much less so in Nature communications.

We thank the Reviewer for having suggested important additional experiments; We believe the points
above, together with (i) new data from additional experiments, (ii) extended in-vivo experiments on
mouse model after 60 days amyloid feeding, and (iii) additional histological analysis on five organs
(spleen, kidney, lung, heart and liver) beside brain, do respond amply and comprehensively to the
criticisms of the Reviewer and hopefully remove the pending ambiguities and clarify further doubts on
the analysis. We also hope these results will help the Reviewer and the scientific community in general
to discard the stereotype between amyloids and their association with the pathological amyloidosis in
the context of digestio: the results show that no detectable toxicity exist from digested food amyloid fibrils
and even that pathological amyloids, when digested, would be no longer cytotoxic as shown by cell
viability on two cell lines exposed to digested A β fibrils (although this would require additional in-vivo
work which is totally out of the scope of this story). With the large evidence in-vitro and in-vivo produced
in the revised version of the study, largely inspired by this and the other two Reviewers, we hope to
have convinced the Reviewer on the robustness of our results and conclusions.

Reviewer #2 (Remarks to the Author):

In their manuscript entitled “Food Amyloid Fibrils as Safe Nutrition Ingredients: In-vitro and In-vivo
Assessment”, Xu, Mezzenga et al. explore the safety aspects of using food protein amyloid fibrils as
potential ingredients for human nutrition. The authors use a vast portfolio of methods, including in vivo
approaches in animal models, to address the central question as to whether potentially pathologic
amyloidogenic core structures remain after passage of fibrillar protein aggregates through the digestive
system. This is obviously an important question in the field of food chemistry as it has to be ruled out
conclusively that remaining amyloid nuclei can exert nanoparticle-related cytotoxicity or even induce
pathological amyloid structures via cross-seeding, before protein amyloids can be considered as food
additives. The authors show that monomeric and fibrillar forms of selected food proteins are digested
into essentially the same, non-toxic oligopeptides and consider both as equally safe, paving the way for
protein fibrils as beneficial food ingredients.

Although the major message is convincingly supported by the experiments overall, the manuscript
appears a bit like compiled in a rush. There are some shortcomings with the description of the methods
used and the presentation of the data obtained, in particular related to the mass spectrometric
experiments, which need to be addressed to improve the quality of the manuscript. Specifically, the
mass spectra in Fig. 3 need scaled y-axes, ideally showing absolute intensity in the same range per
spectra set to allow for semi-quantitative comparison.

We agree with the Reviewer. Following the suggestion, we have added the y-axes in the plots in the
revised version of the manuscript. We have also made edits throughout the manuscript for better clarity
and composition.

While appropriate for the average mass signals in the protein spectra from linear mode, the
monoisotopic mass signals in the peptide spectra from reflector mode should be annotated by masses
with two digits in line with the high resolution/mass accuracy of the latter mode. As signals can in
principle also originate from autolysis/cross-digestion of the enzymes involved (or from CHCA matrix
clusters at least in the low m/z range of the spectrum), the data would be much more convincing if the
signals would be mapped onto the respective substrate protein sequence in the sense of a peptide
mass fingerprint. This can be done via a table comparing calculated and observed masses in the
supplementary information or, more elegantly, by a database search of the peak lists (similar to what is
done later with the LC-MS data).

We totally agree with Reviewer and we are well aware of this point. However, the ratio of enzyme vs
digested protein in the in-vitro part of the study has been intentionally chosen to be as low as possible
(while allowing comprehensive digestion), precisely to minimize autolysis/cross-digestion of the enzyme:
to highlight this, one can use the method of van der Linden et al. Langmuir 2011
(<https://pubs.acs.org/doi/pdf/10.1021/la104797u>) quantifying concentrations based on SDS bands

intensities. We accordingly always find that the ratio (enzyme vs digested protein) is in the order of 1:10
after a careful calculation on the SDS gels, which is comparable to the signal/noise ratio in the MALDI.
Following comments by Reviewer 1, we have now included a part (based on AFM) to digest amyloid
fibrils also at higher enzyme content (please check reply to comment 7 by Reviewer 1).

Furthermore, it is known that the intensities of MALDI are not quantitatively accurate (as opposed to the
highly accurate mass/charge values) and therefore this quantitative analysis, would not be accurate
anyhow. Due to these reasons, we have used a much more accurate LC-MS analysis in the *in vivo*
study, providing a precise picture of the hydrolyzed peptides *in vivo* in the GI tract.

Along the same lines, the signal-to-sequence assignment should be corroborated by mass
spectrometric sequencing of the respective peptides, at least for species explicitly mentioned in the text
like the dominant peaks common/differential in Ig monomer and fibril, respectively. Given the instrument
used for generating the data shown, MALDI-TOF/TOF-MS/MS is accessible to the authors.
In general, it is recommended that this part of the text is proofread by an expert in mass spectrometry.

We appreciate the comment of the reviewer. At the low enzymatic activity used for the *in-vitro* part, we
do know that enzyme autolysis peptide remains of the order of traces, however, that some amyloid may
not be entirely digested; when increasing the enzyme concentration, yet, still within the balance of the
INFOGEST limits, we know that amyloid digestion will be complete, but that we will enter into a
significant enzyme autolysis; therefore, sequencing the MALDI-TOF for the *in-vitro* part will anyhow
suffer of any of the two issues above (or a combination thereof), depending in which enzyme
concentration regime we set the *in-vitro* analysis. Precisely for these reasons, the LCMS sequencing
has been avoided in the *in-vitro* part, but it is an essential part of the *in-vivo* part, for which the choice of
enzyme concentration do no longer become a determinant factor. We hope the Reviewer can understand
this and appreciate instead that what requested is already done for the most prominent, *in-vivo* mouse
study section.

With respect to the second comment, as suggested, we made further improvements in the MALDI-TOF
results description with support of expert scientists. The modifications have been added to the revised
manuscript and the scientist duly acknowledged for the technical support and proofreading.

as there are several issues that need correction/commenting:

- The calculation of the multiply charged species is incorrect as the mass of the proton is neglected; for
example, the doubly charged species is $(18366+2)/2$

We thank the Reviewer for noticing this. We agree and corrected the doubly charged species to be
$[M+2H]^{2+}=(18366+2)/2$ and triply charged species to be $[M+3H]^{3+}=(18366+3)/3$. These corrections have
been updated in the revision.

- It is unusual that the doubly and not the singly charged species is the dominant signal *i* in a MALDI
mass spectrum (see Fig. 3a). This is likely due to the use of CHCA as matrix, which is the most
commonly used matrix for peptides, but not for proteins. The authors may want to consider to comment
on that phenomenon, which may otherwise confuse readers expecting a 'normal' charge distribution

We fully agree with the Reviewer on this comment. A sentence has been added in the method section
"we note that the singly charged protein species is not the dominant signal, and we believe this is due to
the use of CHCA matrix" to clarify this in the revised manuscript.

- Double-check terminology: mass spectroscopy vs. mass spectrometry (correct form); reflection mode
vs. reflector mode (correct form)

We thank the Reviewer for the modification. The terminologies have been corrected in the revised
manuscript.

While the necessary information on MALDI-MS is provided in the Methods section, the description of the
proteomic analysis of the *in vivo* digestion products is absolutely insufficient, particularly in the absence
of any references for the methods used. First, details are needed for the C18 SPE of the
intestinal/serum samples (loading/washing/elution). For serum samples, was the cartridge eluted in a
way that peptides were separated from serum proteins? Or was it a bulk elution of all peptides/proteins
bound to the stationary phase? The latter case may explain why no peptides were detected in the serum
samples. As to the LC-MS conditions, the minimal set of information should include: LC system,
column(s) used, mobile phases, gradient, mass spectrometer type, data acquisition mode,
resolution/AGC targets, mass range. Similarly, information on data processing should include search
parameters (no enzyme setting? Cys-status?) and quantification method (LFQ?). A complete
protein/peptide list extracted from the MaxQuant results must be part of the supplementary information
(an incomplete table like the one shown as Table S3 does not tell anything).

For identification of serum samples, there were many peptides detected in the serum samples, but when
we matched them with β -Ig sequence database (Table S3), we found two peptides that also were found

in the control samples and no peptides was found from lysozyme, This demonstrates two important
things: that the presence of these two oligopeptide sequences is not coming from exogenous β -lg
amyloids diet and at the same time that peptide presence is detectable by this method and thus
absence of relevant peptides is not conditioned by a flaw of the method used.

The following details of LC-MSMS conditions have been added in the revised manuscript:

*LC-MS/MS method was used to separate and identify the peptides in-vivo⁴⁹. MS measurements were*
*performed on a Q ExactiveTM mass spectrometer coupled to Easy nLC (Thermo Fisher Scientific)*
*according to a reported protocol. Specifically, 15 μ L of each sample was injected from the autosampler*
*to Zorbax 300SB-C18 peptide traps (Agilent Technologies, Wilmington, DE), and then separated by*
*liquid chromatography column (15 cm long, 75 μ m I.D., 5 μ m resin, C18-reversed phase column) which*
*was equilibrated by buffer A (0.1% formic acid solution) and then separated with a linear gradient of*
*buffer B (0.1% formic acid in 84% acetonitrile) at a flow rate of 250 nL/min controlled by IntelliFlow*
*technology over 120 min. MS data was acquired using a data-dependent top10 method which*
*dynamically chose the most abundant precursor ions from the survey scan (300–1,800 m/z) for high-*
*energy collisional dissociation (HCD) fragmentation. The target value was determined based on*
*predictive automatic gain control (AGC) and the dynamic exclusion duration was 20 s. Survey scans*
*were acquired at a resolution of 70,000 at m/z 200 and resolution for HCD spectra was set to 17,500 at*
*m/z 200. Survey scans were acquired at a resolution of 70,000 at m/z 200 and HCD spectra resolution*
*was set to 17,500. Normalized collision energy of 27 eV was used while the underfill ratio, which*
*specifies the minimum percentage of target value likely to be reached at maximum fill time, was set at*
*0.1%. peptide recognition mode was enabled throughout the run. MS data were processed using the*
*MaxQuant software (version 1.5.5.1) by UniProt database without specifying enzyme cleavage rules for*
*the processed MGF files. The following search parameters were used: ± 20 ppm for peptide mass*
*tolerance, 0.1 Da for MS/MS tolerance, and 2 for maximum missed cleavage (with an allowance for 2*
*missed cleavages). Variable modification: Oxidation (M). Label-free peptide quantification (LFQ) based*
*on extracted ion chromatograms and spectral counts and validation was performed in the MaxQuant*
*software. The cutoff value of global false discovery rate (FDR) for peptide identification was set to 0.01.*

Another severe shortcoming in data presentation is the statistics. In the text related to Fig. 2 c-f, the
authors claim to couple AFM imaging with “comprehensive statistical analysis”. However, no details on
the statistical test(s) applied can be found in the Methods section, nor are any p-values reported in the
legend/displayed in the figure panels. Figure S8 is a particular bad example in this regard, as it even
does not display error bars and is thus meaningless.

We agree with the Reviewer on this comment. We actually did analyze the morphology (height, length
and total length) of fibrils, gastric digested fibrils, intestinal digested fibrils in a statistical way but we
have not provided enough details. The description of the AFM analysis and results have been
substantially improved in the revised manuscript. Error bars have been added in all the analysis of new
AFM images.

Minor issues (referring to which is hampered as no page and line numbers are provided):

Thanks for the suggestion. We have added the page and line numbers in the revised manuscript and
supporting information.

- General: as the headlines of the Results section only contain the methods applied (mostly as
abbreviations), the manuscript reads a bit like an old-fashioned article from the field of analytical
sciences and does not come with the more modern flow the broad-interest readers are used to in Nature
Communications. The authors may want to consider to use headlines expressing the major findings and
to provide a short summary/conclusion at the end of each section.

As suggested, the headlines of result section have been modified in the revision of manuscript

Along similar lines, the Discussion mainly summarizes the data and is rather weak on what is gained for
the field of food chemistry by the findings presented. Valid for all parts of the text, the authors tend to
use very long sentences (see for example the last sentence of the section ‘AFM and TEM imaging’) and
should consider rephrasing into shorter, separate sentences.

Similar to the last point, these modifications have been made in the revised version of the manuscript.

- The allergenicity plots in Fig. 1d,e and Fig. S2 are hard to read for the non-expert and may need some
more explanation, particularly as it does not become clear from the Methods section which ELISA
assays were used. For example, the authors explicitly refer to the high initial allergenicity of Ig-
monomers in Fig. 1d, but the initial allergenicity of the Ig fibril seems to be about twice as high(?). Here,
readers from outside food chemistry/allergology may need more guidance. In general, it would be
helpful to keep consistent the figure design of Fig. 1b,c/ Fig. 1d,e and their corresponding supplementary

Figures Fig. S1/S2, e.g. red arrows pointing to protein bands, color shading for milk etc. (not explained
in the legend).

We agree with the Reviewer. In the revised version of the manuscript the following sentence was added
to explain the ELISA in the method section:

770as indicated by ELISA assay (Fig. 1d). This is a technique for detecting and quantifying target
macromolecules such as peptides, proteins or antibodies, as antigens via specific immobilization with an
antibody linked to a reporter enzyme. Here, we applied ELISA to detect...

Specifically, the reason for higher initial β -Ig fibril allergenicity over monomer is due to the increase in
concentration of ELISA enzyme binding epitopes present on the fibrils which were concentrated after
dialysis.

- TO and ThT fluorescence assays are shown for Ig fibrils, but only TO for lysozyme fibrils. Is there a
reason for not showing ThT for the latter?

Thanks for pointing out. TO dye has a broader range of pH in fluorescence assay, while ThT dye is
sensitive especially at acidic pH during the gastric digestion and has also been reported to be
dependent on fibril surface charge (<https://doi.org/10.1016/j.jcis.2020.03.075>); therefore we kept the ThT
only as a control test in this fluorescence assay (from text: *These results were further confirmed by*
*Thioflavin T (ThT) fluorescence assay in Fig. S4*).

- What is the size of the scale bars in Fig. 2a? Likely not 100 nm as given in the legend to panel b.

We thank the Reviewer for having noticed this. These scale bars were 1 μ m; these images have
anyhow been removed and replaced by a more comprehensive AFM analysis.

- Results in vivo, end of first paragraph: the authors conclude that the fibrils influenced the metabolism
of *C. elegans*. Isn't it the improved digestibility of the fibrils that is meant?

We thank the Reviewer for the comment. We found the *C. elegans* showed higher relative motilities (and
therefore improved health) while fibrils fed in both cases of β -Ig and lysozyme; although the mechanism
behind is still unclear for us, we have reproduced the results in another set of experiments and
confirmed again the same trends: higher relative motilities / improved health is systematically and
consistently found when *C. elegans* is fed by β -Ig and lysozyme fibrils. The additional dataset is given in
the supporting info as Fig. S17, whereas in the main text it is added the sentence: "We observed higher
relative motilities and improved health when *C. elegans* is fed by β -Ig and lysozyme fibrils (Fig. 5a, Fig.
S17, Table S1)."

Fig. S17. Duplicate experiment of *C. elegans* motility, demonstrating higher relative motilities /
improved health when *C. elegans* is fed with β -Ig and lysozyme fibrils.

- The distribution of monomers/fibrils is monitored after fluorescent labeling with an amine-reactive dye.

This means that labeled lysine residues escape cleavage by trypsin, which is likely the major protease in
pancreatin. This should be discussed.

We thank the Reviewer for this comment. We agree with Reviewer that labelling the lysine may affect
the cleavage from trypsin, which allows us to further note that the fluorescence results on digestion are
likely to be conservative, since fluorescent labelling of lysine may protect it against trypsin cleavage,
partially inhibiting proteases: this implies that unlabeled fibrils are most likely digested in-vivo even to a
larger extent and we believe that the protease is even higher in vivo and more labeled lysine residues
were cleaved by trypsin in the mice study.

We further note the following: 1) most available fluorescent probes are mainly reactive via amines. 2)
the dye chosen is reactive also towards primary amines (10.1016/0731-7085(89)80066-4) which are
present more in abundance in hydrolyzed peptides compared to lysines. 3) the enzymatic cleavage by
trypsin is only partially inhibited as trypsin also cleaves at arginine residues. 4) the presence of other
pancreatic enzymes (chymotrypsin and carboxypeptidase) would also play a part in cleaving other
amino acids.

As this discussion remains substantially qualitative, we refrain from adding it to the revised manuscript.

- SDS-Page vs. SDS-PAGE (correct form)

We thank the Reviewer, and amended it in the revised manuscript.

- AD mice: the mouse model used is not mentioned

The AD mice model is FAD^{4T} mice, and this has been specified in the revised manuscript.

- Methods section: "... were analyzed using the proteomics method." There is no one proteomics
method

We thank the Reviewer for pointing this out. This will now be revised as "...peptides identification
were studied by LC-MS/MS analysis".

- Fig. S9: this reviewer is not an expert in AFM, negative heights on the color scale may need
explanation.

We thank Reviewer for this point. It is common for AFM images showing a color scale from negative to
positive, this is because the AFM height is a relative value depending on the piezo extension in z-
direction and the user can define the absolute value as needed. Normally the top of the substrate is
defined as zero, but the substrate can also sometimes be slightly negative due to the surface roughness
and machinal noise, and thus the full color scale is normally presented.

- Milli-Q water is a brand name, may be replaced by deionized water

We thank the Reviewer and this has been amended it in the revised manuscript.

Reviewer #3 (Remarks to the Author):

The manuscript by Xu et al. examined the safety of two main types of food amyloids, i.e., whey protein
beta lactoglobulin (β -lg) and plasma protein lysozyme. The central strategy of the study was to mimic
the digestion of food amyloids in the gastric and intestinal tracts with the introduction of two enzymes in
pepsin and pancreatin, and to further characterise the products post the enzymatic activities using a
range of analytical techniques of AFM, absorbance, SDS-PAGE, CD, fluorescence, MALDI-MS,
polarised light microscopy as well as SAXS. In all instances, the products were shown convincingly to
be low in containing a cross-beta backbone characteristic to amyloid fibrils and were down to the level of
peptides post digestion. The biocompatibility and hence safety of the amyloid fibrils was validated by
behaviour assays employing C elegans and mice in vivo models.

The analytical (including MALDI-MS) and animal behavior data presented in this study were of top
quality, as expected from this leading group in functional amyloid structure and food protein science. As
a prime goal of this study was to demonstrate the non-toxic nature of food amyloids, the authors are
suggested to complement their analytical-behavior study with additional toxicity data (e.g., ROS,
apoptosis, and immune response using cell culture or the animal tissue) to fully establish the whole
safety parameters for food amyloids.

By including such cellular/tissue toxicity data this study would be more complete in the reviewer's
opinion and could serve as a comprehensive benchmark study for guiding future biological and
environmental applications of functional amyloids.

We agree with the Reviewer and we have carried out additional experiments accordingly. Firstly, we
have carried out additional in-vitro cytotoxicity studies in two human cell lines by exposing them to the
digested amyloid fibrils (β -lg and lysozyme), digested monomers (β -lg and lysozyme) or the digestion

matrix alone, and found that the digested amyloid fibrils nor digested monomers induced a reduction of
cell viability, rather both digested amyloids and monomers increased the livability of the cell compared to
the reference buffer, as expected by the availability of hydrolized peptides servig as nutrients to the cells,
without systematic and significant differences between the digested monomer and amyloid series.

We have also run and added ROS experiments as requested by the reviewer, further reinforcing these
results.

Finally, inspired by Reviewer 1, we further added the positive control of Abeta fibrils digestion on the
same two cell lines, shoing no detectable toxicity form digested pathological fibrils and reinforcing this
part of the study. These results are a major improvement of the manuscript for which we sincerely wish
to thank the Reviewer. The following text and figures were added:

894 **Impact on viability in human cell lines**

To evaluate the biocompatibility of the *in-vitro* digested amyloids, we first characterized their effect on
the viability of two types of human intestinal cell lines (cancer coli-2, Caco2; human colon epithelial cells,
HCEC). Caco2 cells are widely used to study intestinal transport, but have phenotypic characteristics of
cancer cells, whereas HCEC cells are complementary in being immortalized cells with no pathological
features, and therefore an attractive model for testing potential impact on healthy cells⁴¹⁻⁴³. We
characterized cellular responses of exposure to the complex buffer used for the *in-vitro* digestion buffers
(matrix), as well as the effects of digested monomer or digested amyloid from both β -lg and lysozyme
(**Fig. S15**). These assays were performed for fibril and monomer concentrations ranging from 50 to
1,250 μ g/mL, as well as corresponding dilutions of the digestion matrix. When we compared the viability
of both cell types following exposures to digested amyloids vs. digestion matrix, or digested monomers
vs. digestion matrix (**Fig. 3a**), it was apparent that there was no loss of cell viability due to the presence
of the digestion products. In fact, in both cell lines there was growth promotion upon exposure to
digested β -lg and lysozyme fibrils or monomers relative to exposures to the digestion matrix alone (**Fig.**
**4**), consistent with an increase in bioavailable nutritive protein components.

No significant differences were found when comparing the viability of cells upon exposure to digested
monomer vs. digested fibril, in both Caco2 and HCEC cell lines at all concentrations. The highest
concentration of digestion matrix alone caused a strong reduction in cell viability (**Fig S15**), however the
relative influence of the digested fibrils or monomers at any of the concentrations was not associated
with any additional negative impacts (**Fig S15**). To address whether digestion is even capable to
suppress toxicity of pathological amyloids, we additionally tested $A\beta_{42}$ fibrils as pathological amyloid
model. Interestingly, we found that even digested $A\beta_{42}$ fibrils did not reduce cell viability (**Fig S16**). Thus,
the observations regarding overall cell viability are fully consistent with predictions from the digestion
behavior, supporting that on a first-tier cytotoxicity assessment, digested fibrils are as safe as their
digested monomers as there is no detectable cytotoxic effects, and furthermore, that they are well
broken down to biocompatible and nutritive peptides, even in the case of $A\beta_{42}$ fibrils.

These results were further supported by reactive oxidative species (ROS) experiments, as shown in Fig.
4e and 4f, where for both β -lg and lysozyme fibrils and in both Caco2 and HCEC cells, fluorescence
indicating the generated stress had comparable values for digested fibrils and their monomer controls.

Fig 4. | Effect of *in-vitro* digested protein monomers and amyloids on cell viability. The ratio of Caco2 cell viability treated with β -Ilg (a) and lysozyme (b) fibrils and monomers respectively, vs. digestion matrix at the treatment concentration of 50, 250, 500 and 1250 $\mu\text{g/mL}$. The ratio of HCEC cell viability treated with β -Ilg (c) and lysozyme (d) fibrils and monomers respectively, vs. digestion matrix at the treatment concentration of 50, 250, 500 and 1250 $\mu\text{g/mL}$. Reactive oxidative species experiments for e) β -Ilg and f), lysozyme digested fibrils and their monomer controls.

Furthermore and in addition to the *in vitro* characterization, following the reviewer's comment about animal tissues and in line with the requests by Reviewer 1, we extended our original *in vivo* study in a mouse model and have histologically examined five organs plus brain after a 60 day feeding study. These results confirm and reinforce the message that no cross-seeding occurs between the digested food amyloids and pathological or systemic amyloidosis species with all tissues inspected histologically matching tissues of control healthy mice.

Other suggestions/minor issues:

1. Fig. 5 and its relevant text, here the abbreviation "pre-digested" referred to already digested amyloids, in contrast to the other sample of "amyloids", but the term sounded like for amyloids prior to digestion. Consider modifying the abbreviation to avoid confusion.

We agree with the Reviewer. The abbreviation "pre-digested" has been revised as "digested" as suggested, in the revision of manuscript.

2. Fig. S20 was an overview of the biological fate of β -Ig monomer and amyloid fibrils after entering the gastrointestinal tract. This scheme might be better placed in the main text for readability.

We fully agree with the Reviewer's suggestion, and we would very much like to place it in the main text, but we are limited on the amount of figures so that we must prioritize the most valuable dataset on the main text. If the Editor allows it, we happily move it to the main text.

3. Minor language issues, such as in Abstract: "...monomers from (a) technological, nutritional, sensorial, and physical perspective(s)"; in Introduction: " Homo sapiens has (have) been consuming dairy..." "...may reach the bloodstream and nucleating (nucleate) the formation of pathological amyloids by cross-seeding..." "it was therefore chosen as (a) main model in this digestion study"; in Results: "Similarities of the peptide sequences on β -Ig amyloid fibrils after fibrillization with some of the IgE-binding epitopes from earlier studies highly supports (support) that"...

We thank the Reviewer for noting these errors, and have made writing edits throughout to improve clarity and composition. *Homo Sapiens*, however, is the name of the species, it is singular, so the verb was conjugated correctly.

REVIEWER COMMENTS

Reviewer #1 (Remarks to the Author):

In the revised manuscript by Xu et al., the authors have demonstrated the feasibility of beta-lactoglobulin (β -lg) and lysozyme amyloid fibrils as superior food protein supplements. The authors have sufficiently addressed the concerns raised by performing additional experiments that support their claims. The results indicate that food protein amyloid fibrils are at least equally safe as monomeric counterparts and have been validated by multiple in vitro and in vivo experiments.

Overall, the manuscript has improved significantly after the revision, and the revised version can be accepted for publication in Nature Communications.

Reviewer #2 (Remarks to the Author):

The authors have fixed the issues with the scaling of the y-axes in the MALDI mass spectra (new Fig. 2, unfortunately still not in the corresponding Suppl. Fig.) and have addressed that the observed signals are unlikely to be derived from protease autolysis. However, in the way the mass spectral data are presented in the insets, they still do not support the conclusion as claimed by the authors (lines 201-207):

“Interestingly, four dominant peaks (930, 1059, 1130, and 1313 m/z) can be found in the reflector mode spectrum of both digested β -lg monomer and amyloid fibril due to the homologous sequence of β -lg monomer and fibril, while two additional peaks at 1245 Da and 1508 Da were found in the digested β -lg monomer but not on the digested amyloid fibrils. The most important conclusion from these results is that the oligopeptides left over from the digestion of β -lg fibrils are also present in the digested β -lg monomers, and therefore, that residual oligopeptides from digested β -lg fibrils cannot be more toxic than those from the digested homologue β -lg monomers.”

The authors should make use of the high mass accuracy provided by the reflector mode and show the m/z values with at least two digits (maybe in a Suppl. Tab.) so that the reader can judge if signals from two different mass spectra are indeed likely to be the same. Given the central importance of this conclusion, this Reviewer still misses any efforts to assign the mass signals observed to sequences in β -lg, either by mass mapping or ideally by MALDI sequencing of the corresponding peptides.

Also related to the data in new Fig. 2, the issue with the calculation of the m/z of the multiply charged species is addressed in the response letter, but was not properly corrected in the manuscript text.

In the methods section, the authors have now provided details on the LC-MS method used as per request of this Reviewer. Any redundancy to the previous paragraph should be removed. Some of the new details provided raise new questions: first, the authors allow for two missed cleavages although they do the database search “without specifying enzyme cleavage rules” (line 690). This does not seem to make sense. Second, the authors claim that they used “Label-free peptide quantification (LFQ) based on extracted ion chromatograms and spectral counts (line 693)”. The latter method is rather obsolete in our days and they should clearly state which quantification method was used for which experiment in the present study.

As to the requested details on the antigenicity measurements, the authors now provide some general explanation of the ELISA methodology. Why this is certainly helpful for a broad readership, it was not the initial intention of this Reviewer. Rather, it was meant to provide details on which ELISA kits were used as only a manufacturer is mentioned in the methods section, but without any further details.

As to the general appearance of the manuscript, the authors claim that they have modified the headlines of the results section. However, this is not really the case, at least not along the lines of the original suggestion of this reviewer. The idea was to give the results part a more modern flow, instead of an old-school listing of the respective methods applied.

Reviewer #3 (Remarks to the Author):

My main concern for the original manuscript was its relatively light toxicity data alongside the more extensive biophysical and analytical data which intended to offer a comprehensive assessment of functional amyloids as food nutrients. This concern has now been adequately addressed in the revision by the inclusion of Figs. 4, S15-16, the original Figs. S21-22, as well as the new histology data in Figs. 6f&S23 to eliminate the possibility of cross-seeding by functional amyloid for pathogenic Abeta to compromise the safety of food amyloid in vivo.

I have also studied with interest the authors' response to other reviewers' comments and based on the quality of the revision and the quality of the response letter I endorse acceptance of this article by Nature Communications.

Rebuttal letter

Reviewer #1 (Remarks to the Author):

In the revised manuscript by Xu et al., the authors have demonstrated the feasibility of beta-lactoglobulin (β -lg) and lysozyme amyloid fibrils as superior food protein supplements. The authors have sufficiently addressed the concerns raised by performing additional experiments that support their claims. The results indicate that food protein amyloid fibrils are at least equally safe as monomeric counterparts and have been validated by multiple in vitro and in vivo experiments.

Overall, the manuscript has improved significantly after the revision, and the revised version can be accepted for publication in Nature Communications.

We thank the Reviewer for the positive comment.

Reviewer #2 (Remarks to the Author):

The authors have fixed the issues with the scaling of the y-axes in the MALDI mass spectra (new Fig. 2, unfortunately still not in the corresponding Suppl. Fig.) and have addressed that the observed signals are unlikely to be derived from protease autolysis. However, in the way the mass spectral data are presented in the insets, they still do not support the conclusion as claimed by the authors (lines 201-207):

“Interestingly, four dominant peaks (930, 1059, 1130, and 1313 m/z) can be found in the reflector mode spectrum of both digested β -lg monomer and amyloid fibril due to the homologous sequence of β -lg monomer and fibril, while two additional peaks at 1245 Da and 1508 Da were found in the digested β -lg monomer but not on the digested amyloid fibrils. The most important conclusion from these results is that the oligopeptides left over from the digestion of β -lg fibrils are also present in the digested β -lg monomers, and therefore, that residual oligopeptides from digested β -lg fibrils cannot be more toxic than those from the digested homologue β -lg monomers.”

The authors should make use of the high mass accuracy provided by the reflector mode and show the m/z values with at least two digits (maybe in a Suppl. Tab.) so that the reader can judge if signals from two different mass spectra are indeed likely to be the same. Given the central importance of this conclusion, this Reviewer still misses any efforts to assign the mass signals observed to sequences in β -lg, either by mass mapping or ideally by MALDI sequencing of the corresponding peptides.

We are most thankful to the Reviewer for having suggested these further revisions and for having spotted the missed changes in the former revision. We have taken all the needed actions to solve this in full in the present revision. Specifically, we have:

- 1. Fixed the supporting figure S8 as requested (scaling of y axis),
- 2. Amended the m/z numbers for the monomer in the main text,
- 3. Reported to the second digit the m/z values from the reflector mode as a new Table S1,
- 4. Mapped the important MALDI m/z values via the Peptide Mapping Fingerprinting (PMF) to strengthen the discussion as suggested by the Reviewer,
- 5. Amended the text discussing the MALDI section accordingly, in order to compare comprehensively digested monomers vs digested fibrils.

As the Reviewer will note, the amended discussion further strengthens our conclusions: we are most thankful to the reviewer for having allowed us to make this further improvement in the manuscript.

Also related to the data in new Fig. 2, the issue with the calculation of the m/z of the multiply charged species is addressed in the response letter, but was not properly corrected in the manuscript text.

Thank you for spotting this, which has now been duly corrected in the revised manuscript.

In the methods section, the authors have now provided details on the LC-MS method used as per request of this Reviewer. Any redundancy to the previous paragraph should be removed. Some of the new details provided raise new questions: first, the authors allow for two missed cleavages although they do the database search “without specifying enzyme cleavage rules” (line 690). This does not seem to make sense. Second, the authors claim that they used “Label-free peptide quantification (LFQ) based on extracted ion chromatograms and spectral counts (line 693)”. The latter method is rather obsolete in our days and they should clearly state which quantification method was used for which experiment in the present study.

We are most thankful to the Reviewer for this comment. The redundancy part of the previous paragraph has been removed in the revised manuscript. Regarding the new details of the provided LC-MS/MS method, the “and 2 for maximum missed cleavage (with an allowance for 2 missed cleavages)” has been deleted. In addition, the in-vivo digested products of β -lg/lysozyme amyloid fibrils and monomer and their peptides identification in small intestinal tracts, colon and serum were studied by LC-MS/MS analysis, and the database search of peptides identification by LC-MS/MS was performed using the Label-free peptide quantification (LFQ) intensity in the MaxQuant software, which is based on extracted ion chromatograms. These two modifications have been revised in the manuscript accordingly.

As to the requested details on the antigenicity measurements, the authors now provide some general explanation of the ELISA methodology. Why this is certainly helpful for a broad readership, it was not the initial intention of this Reviewer. Rather, it was meant to provide details on which ELISA kits were used as only a manufacturer is mentioned in the methods section, but without any further details.

We thank Reviewer for this comment. In the revised version of the manuscript the following sentence was added to provide details of the ELISA kit in the method section: “The antigenicity of all samples was evaluated with ELISA assays using polyclonal antibodies quantification kits (AgraQuant Beta-Lactoglobulin and AgraQuant Lysozyme) purchased from Romer Labs Deutschland GmbH, Germany.” We have enquired with the company and these are all the info we are allowed to disclose about these commercial kits. Further details are proprietary to the company.

As to the general appearance of the manuscript, the authors claim that they have modified the headlines of the results section. However, this is not really the case, at least not along the lines of the original suggestion of this reviewer. The idea was to give the results part a more modern flow, instead of an old-school listing of the respective methods applied.

We appreciate this comment of the Reviewer. In the revised version, we did try to compromise between the request of the Reviewer for a modern flow and the need of providing a comprehensive view on the massive set of experimental methodologies used in the manuscript. As this relates more to style than scientific content, and since this will further re-edited at the editorial level, we hope the Reviewer can accept the current setting.

Reviewer #3 (Remarks to the Author):

My main concern for the original manuscript was its relatively light toxicity data alongside the more extensive biophysical and analytical data which intended to offer a comprehensive assessment of functional amyloids as food nutrients. This concern has now been adequately addressed in the revision by the inclusion of Figs. 4, S15-16, the original Figs. S21-22, as well as the new histology data in Figs. 6f&S23 to eliminate the possibility of cross-seeding by functional amyloid for pathogenic Abeta to compromise the safety of food amyloid in vivo.

I have also studied with interest the authors' response to other reviewers' comments and based on the quality of the revision and the quality of the response letter I endorse acceptance of this article by Nature Communications.

We thank the Reviewer for the comment and for appreciating the effort we made in the revision.